# Random Policy Enables In-Context Reinforcement Learning within Trust Horizons

**Weiqin Chen** *chenw18@rpi.edu*
*Department of Electrical, Computer, and Systems Engineering, Rensselaer Polytechnic Institute*

**Santiago Paternain** *paters@rpi.edu*
*Department of Electrical, Computer, and Systems Engineering, Rensselaer Polytechnic Institute*

**Reviewed on OpenReview:** *https://openreview.net/forum?id=mAiMKnr9r5*

## Abstract

Pretrained foundation models (FMs) have exhibited extraordinary in-context learning performance, allowing zero-shot (or few-shot) generalization to new environments/tasks not encountered during the pretraining. In the case of reinforcement learning (RL), in-context RL (ICRL) emerges when pretraining FMs on decision-making problems in an autoregressive-supervised manner. Nevertheless, the current state-of-the-art ICRL algorithms, such as Algorithm Distillation, Decision Pretrained Transformer and Decision Importance Transformer, impose stringent requirements on the pretraining dataset concerning the behavior (source) policies, context information, and action labels, etc. Notably, these algorithms either demand optimal policies or require varying degrees of well-trained behavior policies for all pretraining environments. This significantly hinders the application of ICRL to real-world scenarios, where acquiring optimal or well-trained policies for a substantial volume of real-world training environments can be prohibitively expensive or even intractable. To overcome this challenge, we introduce a novel approach, termed State-Action Distillation (SAD), that allows to generate an effective pretraining dataset guided solely by random policies. In particular, SAD selects query states and corresponding action labels by distilling the outstanding state-action pairs from the entire state and action spaces by using random policies within a trust horizon, and then inherits the classical autoregressive-supervised mechanism during the pretraining. To the best of our knowledge, this is the first work that enables effective ICRL under (e.g., uniform) random policies and random contexts. We also establish the quantitative analysis of the trustworthiness as well as the performance guarantees of our SAD approach. Moreover, our empirical results across multiple popular ICRL benchmark environments demonstrate that, on average, SAD outperforms the best baseline by 236.3% in the offline evaluation and by 135.2% in the online evaluation.

## 1 Introduction

Pretrained foundation models (FMs) have demonstrated promising performance across a wide variety of domains in artificial intelligence including natural language processing (NLP) (Devlin, 2018; Radford, 2018; Radford et al., 2019; Brown, 2020), computer vision (CV) (Yuan et al., 2021; Sammani et al., 2022; Ma et al., 2023; Chen et al., 2024b), and sequential decision-making (Chen et al., 2021; Janner et al., 2021; Xu et al., 2022b; Yang et al., 2023; Light et al., 2024a;b). This success is attributed to FMs' impressive capability of in-context learning (Dong et al., 2022; Li et al., 2023; Wei et al., 2023; Wies et al., 2024) which refers to the ability to infer and understand the new (unseen) tasks provided with the context information (or prompt) and without model parameters updates. Recently, in-context reinforcement learning (ICRL) (Laskin et al., 2022; Grigsby et al., 2023; Lin et al., 2023; Sinii et al., 2023; Zisman et al., 2023; Lee et al., 2024; Lu et al., 2024; Wang et al., 2024; Dong et al., 2024) has emerged when FMs are pretrained on sequential decision-making

problems. Whereas FMs use texts as the context/prompt in NLP, ICRL treats the state-action-reward tuples as the contextual information for decision-making.

However, the current state-of-the-art (SOTA) ICRL algorithms impose strict requirements on the pretraining datasets. More specifically, Algorithm Distillation (AD) (Laskin et al., 2022) requires the context to contain the complete learning history (from the initial policy to the final-trained policy) of the source (or behavior) RL algorithm for all pretraining environments. In addition, AD requires environments to have short episodes, allowing the context to capture cross-episodic information. This enables AD to learn the improvement operator of the source RL algorithm. Conversely, Decision Pretrained Transformer (DPT) (Lee et al., 2024) partially relaxes the requirement on the context, permitting it to be gathered by random policies and without needing to adhere to the transition dynamics. Nevertheless, DPT necessitates the optimal policy to label an optimal action for any randomly sampled query state across all pretraining environments. To explore the feasibility of ICRL in the absence of optimal policies, Decision Importance Transformer (DIT) (Dong et al., 2024) proposes to leverage the observed state-action pairs in the context data as query states and corresponding action labels. Each state-action pair within the context is assigned a weight in the training process. This weight is proportional to the return-to-go of the pair. Thus, DIT prioritizes the training on high-return pairs. Despite not demanding optimal policies, DIT still requires a substantial context dataset to comprehensively cover all state-action pairs from the state and action spaces. Furthermore, DIT still mandates that more than 30% of the context data comes from well-trained policies to ensure the coverage of good action labels, and the context should originate from a complete episode.

Notably, acquiring either optimal policies or well-trained policies across a multitude of pretraining environments in real-world scenarios can be prohibitively expensive or even intractable. On the other hand, the transition data available in real-world problems—collected as the context—may not originate from a complete episode. These stringent requirements on the pretraining dataset of the SOTA ICRL algorithms severely limit their practical applications to the real world, especially for those where the transition data exhibits high variance to train an effective policy. This work thus aims to relax these requirements by considering an untrained random policy. Although the random policy cannot solve all decision-making problems, it is worth noting that the random policy and the optimal policy select the same optimal actions in certain problems, e.g., the grid world navigation problem (Laskin et al., 2022; Lee et al., 2024; Dong et al., 2024) where the reward is received only upon reaching the unique goal. In these problems, the optimal action induced by both policies corresponds to navigating to the goal as quickly as possible. Consequently, this paper centers on the ICRL that operates without the need for optimal (or any degree of well-trained) policies or episodic context, placing its emphasis on the scenarios under (e.g., uniform) random policies and random contexts only. Our applicable domains in this work focus on the multi-armed bandits and the grid world navigation problems with a single sparse reward received only upon achieving the unique goal.

## 1.1 Main Contributions

The main contributions of this work are summarized as follows.

- We propose a novel approach termed State-Action Distillation (SAD) to generate the pretraining dataset of ICRL under random policies. Notably, SAD distills the outstanding state-action pairs over the entire state and action spaces for the query states and corresponding action labels (refer to Figure 1), by executing all possible actions under the random policies within a trust horizon.

- To the best of our knowledge, SAD stands as the first method that enables effective ICRL under (e.g., uniform) random policies and random contexts.

- We establish the quantitative analysis of the trustworthiness as well as the performance guarantees of our SAD approach. We substantiate the efficacy of SAD by empirical results on several popular ICRL benchmark environments. On average, SAD significantly outperforms all existing SOTA ICRL algorithms. More concretely, SAD surpasses the best baseline by 236.3% in the offline evaluation and by 135.2% in the online evaluation.

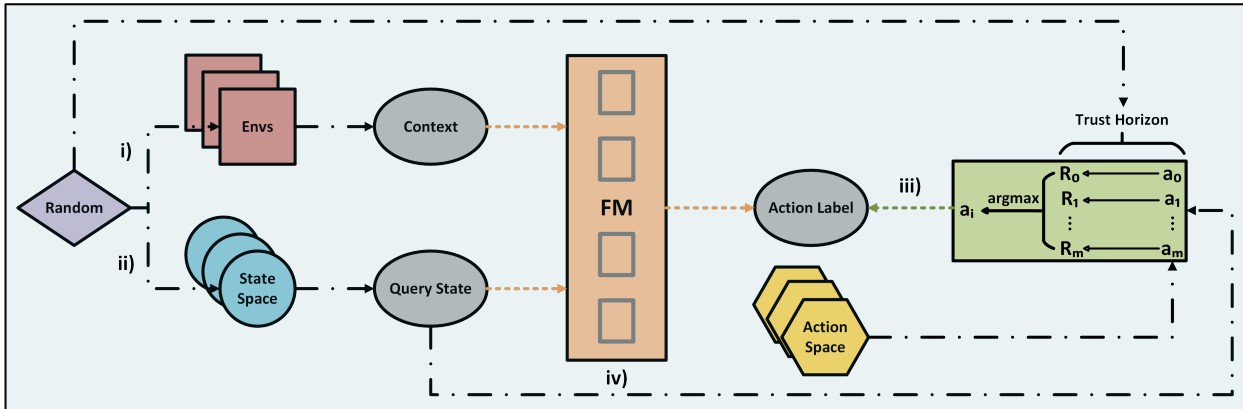

Figure 1: Schematic of the State-Action Distillation approach: i) Collecting the context by using the random policy to interact with pretraining environments. ii) Sampling a query state randomly from the state space. iii) Starting from the query state and any action in action space, running trust horizons under the random policy, and distilling the action label by the action that yields the maximal return. iv) Pretraining foundation models in a supervised mechanism, which predicts the action label given the context and query state.

## 2 Related Work

### 2.1 Offline Reinforcement Learning

In contrast to the unlimited interactions with the environment in online RL, offline RL seeks to learn optimal policies from a pre-collected and static dataset (Fujimoto et al., 2019; Levine et al., 2020; Kumar et al., 2020; Kostrikov et al., 2021; Chen et al., 2024a). One of the critical challenges in offline RL is with bootstrapping from out-of-distribution (OOD) actions (Levine et al., 2020; Kumar et al., 2020; Xu et al., 2022a; Liu et al., 2024) due to the mismatch between the behavior policies and the learned policies. To address this issue, the current SOTA offline RL algorithms propose to update pessimistically by either adding a regularization or underestimating the Q-value of OOD actions.

### 2.2 Autoregressive-Supervised Decision Making

In addition to the traditional offline RL methods, autoregressive-supervised mechanisms based on the transformer architecture (Vaswani, 2017) have been successfully applied to offline decision making domains by their powerful capability in sequential modeling. The pioneering work in the autoregressive-supervised decision making is the Decision Transformer (DT) (Chen et al., 2021). DT autoregressively models the sequence of actions from the historical offline data conditioned on the sequence of returns in the history. During the inference, the trained model can be queried based on pre-defined target returns, allowing it to generate actions aligned with the target returns. The subsequent works such as Multi-Game Decision Transformer (MGDT) (Lee et al., 2022) and Gato (Reed et al., 2022) have exhibited the success of the autoregressive-supervised mechanisms in learning multi-task policies by fine-tuning or leveraging expert demonstrations in the downstream tasks.

### 2.3 In-Context Reinforcement Learning

However, both traditional offline RL and autoregressive-supervised decision making mechanisms suffer from the poor zero-shot generalization and in-context learning capabilities to new environments, as neither can improve the policy, with a fixed trained model, in context by trial and error. In-context reinforcement learning (ICRL) aims to pretrain a transformer-based FM, such as GPT2 (Radford et al., 2019), across a wide range of pretraining environments. During the evaluation (or inference), the pretrained model can directly infer the unseen environment and learn in-context without the need for updating model parameters. The SOTA ICRL algorithms including AD (Laskin et al., 2022), DPT (Lee et al., 2024) and DIT (Dong et al.,

2024) have demonstrated the potential of the ICRL framework. Nevertheless, each of these methods imposes distinct yet strict requirements on the pretraining dataset e.g., requiring well-trained (or even optimal) behavior policies, the context to be episodic and/or substantial, which significantly restrict their practicality in real-world applications. Accordingly, mastering and executing ICRL under (e.g., uniform) random policies and random contexts remains a crucial direction and a critical challenge.

## 3 In-Context Reinforcement Learning

This section introduces the background of ICRL mechanisms and three SOTA ICRL algorithms. We start by presenting the preliminaries of ICRL.

### 3.1 Preliminaries

RL problems are generally formulated as Markov Decision Processes (MDPs) (Sutton, 2018). An MDP can be represented by a tuple $\tau = (S, A, R, P, \rho)$, where $S$ and $A$ denote finite state and action spaces, $R : S \times A \to \mathbb{R}$ denotes the reward function that evaluates the quality of the action, $P : S \times A \times S \to [0, 1]$ denotes the transition probability that describes the dynamics of the system, and $\rho : S \to [0, 1]$ denotes the initial state distribution.

A policy $\pi$ defines a mapping from states to probability distributions over actions, providing a strategy that guides the agent in the decision making. The agent interacts with the environment following the policy $\pi$ and the transition dynamics of the system, and then generates an episode of the transition data $(s_0, a_0, r_0, \cdots)$. The performance measure $J(\pi)$ is defined by the expected discounted cumulative reward under the policy $\pi$

$$J(\pi) = \mathbb{E}_{s_0 \sim \rho, a_t \sim \pi(\cdot|s_t), s_{t+1} \sim P(\cdot|s_t, a_t)} \left[ \sum_{t=0}^{\infty} \gamma^t r_t \right]. \tag{1}$$

The goal of RL is to find an optimal policy $\pi^*$ that maximizes $J(\pi)$. It is crucial to recognize that $\pi^*$ often varies across different MDPs (environments). Thus, the optimal policy for standard RL must be re-learned each time a new environment is encountered. Under this circumstance, ICRL proposes to pretrain a FM on a wide variety of pretraining environments, and then deploy it in the *unseen* test environments without updating parameters in the pretrained model, i.e., zero-shot generalization (Sohn et al., 2018; Mazoure et al., 2022; Zisselman et al., 2023; Kirk et al., 2023).

### 3.2 Supervised Pretraining Mechanism

In this subsection, we introduce the methodology behind ICRL–a supervised pretraining mechanism. Consider two distributions over environments $\mathcal{T}_{\text{train}}$ and $\mathcal{T}_{\text{test}}$ for pretraining and test (evaluation), respectively. Each environment, along with its corresponding MDP $\tau$, can be regarded as an instance drawn from the environment distributions, where each environment may exhibit distinct reward functions and transition dynamics. Given an environment $\tau$, a context/prompt $\mathcal{C} = \{s_i, a_i, r_i, s_i'\}_{i \in [n]}$ refers to a sample from a pretraining context dataset $D_{\text{train}}(\cdot \mid \tau)$, i.e., $\mathcal{C} \sim D_{\text{train}}(\cdot \mid \tau)$, which are collected through interactions between a behavior policy and the environment $\tau$ (see e.g., Algorithm 1). Notably, $D_{\text{train}}(\cdot \mid \tau)$ contains the contextual information regarding the environment $\tau$. We next consider a query state distribution $D_q^\tau$ and a label policy that maps the query state to the distribution of the action label, i.e., $\pi_l : S \to \Delta_{a_l}(A)$, where $\Delta_{a_l}(\cdot)$ denotes the probability distribution and $a_l$ denotes the prediction/output goal of the FM. Then, the joint distribution over the environment $\tau$, context $\mathcal{C}$, query state $s_q$, and action label $a_l$ is given by

$$P_{\text{train}}(\tau, \mathcal{C}, s_q, a_l) = \mathcal{T}_{\text{train}}(\tau) \cdot D_{\text{train}}(\mathcal{C}|\tau) \cdot D_q^\tau \cdot \pi_l(a_l|s_q). \tag{2}$$

ICRL follows a supervised pretraining mechanism. More concretely, a FM with parameter $\theta$ (denoted by $\mathcal{F}_\theta : \mathbb{C} \times S \to \Delta(a_l)$) is pretrained to predict the action label $a_l$ given the context $\mathcal{C}$ and query state $s_q$. To do so, the current literature (Laskin et al., 2022; Lee et al., 2024; Dong et al., 2024) often considers the following objective function

$$\theta^* = \arg\min_\theta \mathbb{E}_{P_{\text{train}}} \left[ l \left( \mathcal{F}_\theta(\cdot \mid \mathcal{C}, s_q), a_l \right) \right], \tag{3}$$

where $l(\cdot, \cdot)$ represents the loss function, for instance, negative log-likelihood (NLL) for discrete-action problems and mean square error (MSE) for continuous-action scenarios. We note that while the current SOTA ICRL algorithms (AD, DPT, and DIT) adhere to a common objective function equation 3, they differ significantly in constructing the context, query state, and action label. Furthermore, it is important to highlight that selecting appropriate action labels in existing ICRL algorithms can be prohibitively expensive. We discuss this in more detail for each of the ICRL baselines that follow.

**Algorithm Distillation.** Instead of learning an optimal policy for a specific environment, AD proposes to learn a RL algorithm itself across a wide range of environments. This is, the FM in AD is pretrained to imitate the source (or behavior) algorithm over the pretraining environment distribution $\mathcal{T}_{\text{train}}$. In general, AD demands a well-trained source algorithm with its complete learning history (from the initial policy to the final-trained policy). Additionally, AD is restricted to the environments with short episode length, as the context must capture cross-episode information of the source algorithm, while standard transformers often have limited context length. In terms of the objective function equation 3, AD takes the state at the time step $t$ as the query state, $a_t$ from the source algorithm as the action label, and the episodic history data $(s_0, a_0, r_0, \cdots, s_{t-1}, a_{t-1}, r_{t-1})$ to be the context.

**Decision Pretrained Transformer.** Instead of being stringent on the context and the environment itself, DPT handles the context and query state in a more general manner. Specifically, in terms of the objective function equation 3, DPT can consider a random collection of transitions as the context $\mathcal{C}$, a random query state $s_q$ drawn from $D_q$, and an optimal action label corresponding to $s_q$. Despite less requirements on the context, DPT necessitates access to optimal policies for the optimal action labels in all pretraining environments, which may not be available in real-world applications.

**Decision Importance Transformer.** DIT proposes to learn ICRL without optimal action labels, following the same supervised pretraining mechanism as DPT. To that end, DIT chooses to consider every possible state and action in the context as the query state and corresponding action label. It is important to point out that DIT requires a (partially) complete episode to form the context, enabling the computation of a return-to-go for each state-action pair. By mapping the return-to-go to a weight that reflects the quality of each state-action pair, DIT can pretrain the FM using the DPT structure, augmented by the weight assigned to each query state and action label. In other words, DIT prioritizes the training on high-return pairs. Notably, DIT still mandates that more than 30% of the context data comes from well-trained policies to ensure the coverage of good action labels.

To summarize, it is worth highlighting that all these SOTA ICRL algorithms necessitate varying degrees of well-trained, or even optimal policies during the pretraining phase. However, obtaining such policies for real-world applications is often prohibitive, as it demands extensive training across a vast number of real-world environments. This challenge becomes even more pronounced in the domains where the transition data exhibits high variance to train an effective policy. Thus, executing ICRL under (e.g., uniform) random policies and random contexts is crucial for enabling the practical application of ICRL in the real-world.

## 4 State-Action Distillation

In this section, we propose the State-Action Distillation (SAD), an approach for generating the pretraining dataset for ICRL under random policies and random contexts (see Figure 1).

As indicated in equation 2, the pretraining data consists of the context, the query state and the action label. We start by introducing the generation of the context under random policies in SAD (refer to Algorithm 1). It is important to highlight that the context is collected through interactions with the environment under any given random policy (e.g., uniform random policy), and notably, the context not necessarily originates from a complete episode. These benefits make SAD potentially well-suited for ICRL's real-world applications with random transition data only. Having collected the random context, we are now in the stage of collecting query states and corresponding action labels for the pretraining of FM under the random policy.

---

**Algorithm 1** Collecting Contexts under Random Policy

---

1: **Require:** Random policy $\pi$, context horizon length $T$, state space $S$, environment $\tau$, empty context $\mathcal{C} = \emptyset$
2: **for** $t$ in $[T]$ **do**
3:     Sample a state $s \sim S$ and an action $a \sim \pi(\cdot|s)$
4:     Collect $(r, s')$ by executing action $a$ in the environment $\tau$
5:     Add $(s, a, r, s')$ to $\mathcal{C}$
6: **end for**
7: **Return** $\mathcal{C}$

---

We proceed by recalling that DIT prioritizes the training on high-return pairs from the context data collected. To that end, DIT assigns a weight $w$ to the loss function during the pretraining phase that is proportional to the return-to-go, i.e., $w(s_t, a_t) \propto \sum_{t'=t}^{T} \gamma^{t'-t} r_{t'}$. Nonetheless, we acknowledge that DIT may not explore to train on good state-action pairs under the random policy for two reasons: ($i$) DIT solely considers to train on the state-action pairs that are observed in the context, which now contains limited and, more importantly, random transition data. ($ii$) Even for the state-action pairs in the collected context, the return-to-go does not necessarily prioritize the optimal pair but rather promotes the pair with high immediate reward, as the discount factor applies starting from the current time step with a horizon of $(T - t + 1)$ only, instead of $T$. This issue becomes even more critical in the problems with sparse rewards.

Under this circumstance, our SAD approach advocates for distilling the outstanding query states and action labels by searching across the entire state and action spaces under the random policy. Before proceeding, we recall the definition of the optimal action in the problems of multi-armed bandit (MAB) and MDP. For any query state $s_q$, the optimal action for $s_q$ corresponds to the action that maximizes the optimal Q-function

$$a^*_{\text{MAB}}(s_q) \triangleq \arg\max_{a \in A} \underbrace{\mathbb{E}\left[r(s_q, a)\right]}_{Q^*_{\text{MAB}}(s_q, a)}, \quad a^*_{\text{MDP}}(s_q) \triangleq \arg\max_{a \in A} \underbrace{\mathbb{E}_{\pi^*}\left[\sum_{t=0}^{\infty} \gamma^t r_t | s_0 = s_q, a_0 = a\right]}_{Q^*_{\text{MDP}}(s_q, a)}, \tag{4}$$

where $s_q$ in the MAB problem refers to the singleton state of bandits, and $\pi^*$ denotes the optimal policy in the MDP. However, both $a^*_{\text{MAB}}$ and $a^*_{\text{MDP}}$ are intractable to obtain in our problem of interest for two reasons: ($i$) computing the expectation in the Q-function demands to sample infinite episodes; ($ii$) one can only have access to the random policy, instead of $\pi^*$. Therefore, we instead consider ($i$) stochastic approximation that uses the average as the unbiased estimate of the expectation due to the law of large numbers; ($ii$) maximizing the episodic return under the random policy.

### 4.1 Trustworthiness of the Random Policy

Subsequently, the crucial question arises: **when can we trust the random policy?** We claim: *The random policy is probabilistically trustworthy for the MAB and MDP problems within a trust horizon.* We formalize this claim for MAB and MDP in this subsection, which relies on the following assumptions.

**Assumption 1.** *The absolute value of the reward $r(s, a)$ is bounded by a positive constant $B$ for all state-action pairs in the MAB and MDP, i.e., $|r(s, a)| \leq B, \forall (s, a) \in S \times A$.*

Note that Assumption 1 is common in the literature (Azar et al., 2017; Wei et al., 2020; Zhang et al., 2021). In particular, in the case of finite state-action spaces, it is always possible to design the reward to avoid the possibility of being unbounded.

**Assumption 2.** *Given a random policy $\pi$, assume that*

$$\arg\max_{a \in A} Q^\pi_{MDP}(s_q, a) = \arg\max_{a \in A} Q^*_{MDP}(s_q, a), \forall s_q \in S. \tag{5}$$

It is worth highlighting first that Assumption 2 may not hold universally for all MDP problems. In fact, it is unlikely that a random policy and the optimal policy select the same optimal actions for all problems. Nevertheless, we acknowledge that Assumption 2 does hold in the MDP problems like the grid world navigation

where the reward is received only upon achieving the unique goal (Laskin et al., 2022; Lee et al., 2024; Dong et al., 2024), which fall into the applicable domains of this work. We visualize in Figure 2 a single-dimensional grid world navigation problem as an example. In particular, the action derived from maximizing the return under the random policy becomes equivalent to that guided by maximizing the return under the optimal policy, as the maximal return induced by both policies corresponds to navigating to the goal as quickly as possible. We formalize this in Proposition 1 and provide a theoretical proof and empirical validation (refer to Appendix A.2). Although Assumption 2 holds for the grid world navigation problems considered in this work, it is crucial to point out that Assumption 2 may not hold for all grid world navigation problems. To this end, we further discuss the validity of Assumption 2 for the grid world navigation with two sparse rewards (see Appendix A.3), and demonstrate that the validity of Assumption 2 depends on problem-specific factors such as the rewards and the discount factor $\gamma$.

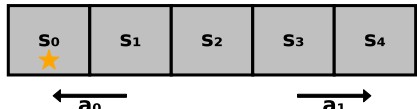

Figure 2: A single-dimensional grid world MDP comprising five states $\{s_0, s_1, s_2, s_3, s_4\}$, where $s_0$ represents the goal state (golden star). The environment offers two possible actions: $a_0$ (go left), and $a_1$ (go right). Any transitions that would result in (left or right) boundary crossing will be confined to the current position. The reward structure is sparse, with a value of 1 received solely upon reaching the unique goal state $s_0$ and a value of 0 otherwise. We consider an infinite time horizon with a discounter factor $\gamma$.

Having introduced the necessary assumptions, we are in the stage of investigating the trustworthiness of the random policy in both MAB and MDP problems. To proceed, we rely on the following two definitions.

**Definition 1** (MAB). *Denote by $s_q$ and $a^*$ the singleton state and optimal arm in the MAB problem. Consider a random policy $\pi$. Suppose that each arm $a$ has been selected $N_a$ times ($N_a \in \mathbb{N}_+$) under $\pi$. The trustworthiness of the random policy in the the MAB problems represents the probability of selecting the optimal arm under the random policy, i.e.,*

$$\frac{1}{N_{a^*}} \sum_{i=1}^{N_{a^*}} r_i(s_q, a^*) \geq \max_{a \in A \setminus \{a^*\}} \frac{1}{N_a} \sum_{i=1}^{N_a} r_i(s_q, a). \tag{6}$$

*The corresponding trust horizon denotes the minimal number of selections over all arms, i.e., $N \triangleq \min_{a \in A} N_a$ that guarantees such trustworthiness.*

Definition 1 implies by the law of large numbers that $N \to \infty$ serves as a trust horizon, ensuring the selection of the optimal arm with probability one. In practice, achieving large trustworthiness requires a sufficiently large trust horizon. The intuition is that a large enough horizon approximates the infinite step problem. The same intuition holds for the MDP problems. We formalize this concept in the next definition, which builds upon the Q-function of the finite-horizon MDP

$$Q_{\text{MDP}}^{\pi,N}(s_q, a) = \mathbb{E}_\pi \left[ \sum_{t=0}^N \gamma^t r(s_t, a_t) | s_0 = s_q, a_0 = a \right], \forall a \in A, \tag{7}$$

$$\hat{Q}_{\text{MDP}}^{\pi,N}(s_q, a) = \frac{1}{N_{\text{ep}}} \sum_{i=1}^{N_{\text{ep}}} \sum_{t=0}^N \left( \gamma^t r(s_t, a_t) | s_0 = s_q, a_0 = a, \pi \right), \forall a \in A, \tag{8}$$

where $\hat{Q}_{\text{MDP}}^{\pi,N}(s_q, a)$ is an unbiased estimate of $Q_{\text{MDP}}^{\pi,N}(s_q, a)$ and $N_{\text{ep}}$ denotes the number of episodes.

**Definition 2** (MDP). *Denote by $a^*$ the optimal action given a query state $s_q$. Consider a random policy $\pi$ as well as its Q-function $Q_{MDP}^\pi$. The trustworthiness of the random policy in the the MDP problems represents the probability of selecting the optimal action for any query state in the state space, i.e.,*

$$\hat{Q}_{MDP}^{\pi,N}(s_q, a^*) \geq \max_{a \in A \setminus \{a^*\}} \hat{Q}_{MDP}^{\pi,N}(s_q, a), \forall s_q \in S. \tag{9}$$

*The corresponding trust horizon denotes the minimal horizon length of the MDP such that*

$$Q_{MDP}^{\pi,N}(s_q, a^*) \geq \max_{a \in A \setminus \{a^*\}} Q_{MDP}^{\pi,N}(s_q, a), \ \forall s_q \in S. \tag{10}$$

In both Definitions 1 and 2, the trustworthiness refers to the probability of selecting the optimal action. While the trust horizon in both cases approximates the infinite-horizon problem, they show subtle differences in the relationship to the trustworthiness. In the MAB problem, which consists of a single episode, the trustworthiness depends on the trust horizon only. However, in the MDP, computing $Q_{\text{MDP}}^{\pi,N}(s_q, a)$ is generally intractable, thus requiring the use of an unbiased estimate (as defined in equation 8), which depends on the number of episodes $N_{\text{ep}}$. Consequently, the trustworthiness in the MDP setting is influenced by both the trust horizon and the number of episodes. Next, we formally quantify the relationship between the trustworthiness and the trust horizon for the MAB and MDP problems.

**Theorem 1** (MAB). *Let Assumption 1 hold. The random policy is at least $(1 - \delta)$-trustworthy as in Definition 1, when the trust horizon $N$ satisfies*

$$N \geq \frac{8B^2}{\left( \mathbb{E}[r(s_q, a^*)] - \max_{a \in A \setminus \{a^*\}} \mathbb{E}[r(s_q, a)] \right)^2} \log \left( \frac{1 + \sqrt{1 - \delta}}{\delta} \right). \tag{11}$$

*Proof.* See Appendix A.1. □

Theorem 1 implies that the trust horizon $N$ quantifies the trustworthiness of the decision making under the random policy $\pi$ for MAB problems. Indeed, a larger $N$ implies a higher probability (smaller $\delta$) that the average reward of the optimal arm under $\pi$ exceeds that of the next-best arm, therefore, making a more reliable decision. We substantiate this claim by empirical evidence (depicted in Figure 6(a)). In the practical implementation, we simply execute the MAB under the random policy $\pi$ until every action in the action space $A$ selected at least $N$ times. Subsequently, we select the action with the maximal average reward as the action label. The detailed procedure for collecting such action labels in MAB is outlined in Algorithm 2.

---

**Algorithm 2** Collecting Query States and Action Labels under Random Policy (MAB)

1: **Require:** Random policy $\pi$, singleton query state $s_q$, action space $A$, environment $\tau$, empty average reward list $L_r$, trust horizon $N$
2: Execute the MAB in $\tau$ under the random policy $\pi$ until every action in $A$ selected at least $N$ times
3: **for** $a$ in $[A]$ **do**
4:     Record the average reward associated with the action $a$ in the history, and add it to $L_r$
5: **end for**
6: Obtain $a_l = A(\arg\max(L_r))$
7: **Return** $(s_q, a_l)$

---

**Theorem 2** (MDP). *Let Assumptions 1 and 2 hold. Define $\kappa = \min_{s_q \in S} \left( Q_{MDP}^{\pi}(s_q, a^*) - \max_{a \in A \setminus \{a^*\}} Q_{MDP}^{\pi}(s_q, a) \right).$ Consider the trust horizon $N > \log_{\gamma} (\kappa(1 - \gamma)/(2B)) - 1$. The random policy is at least $(1 - \delta)$-trustworthy as in Definition 2, when the number of episodes $N_{ep}$ satisfies*

$$N_{ep} \geq \underbrace{\frac{2 \left( 1 - \gamma^{N+1} \right)^2}{\left( \kappa \left( 1 - \gamma \right) / (2B) - \gamma^{N+1} \right)^2}}_{G_1} \log \left( \frac{1 + \sqrt{1 - \delta}}{\delta} \right). \tag{12}$$

*Proof.* See Appendix A.4. □

Theorem 2 implies that the trust horizon $N$ and the number of episodes $N_{\text{ep}}$ quantify the trustworthiness of the decision making under the random policy $\pi$ for MDP problems. Notice that $G_1$ in equation 12 is monotonically decreasing with respect to the trust horizon $N$ when $N > \log_{\gamma} (\kappa(1 - \gamma)/(2B)) - 1$ (see

Lemma 2 in Appendix A.5). Thus, with a fixed number of episodes, a larger $N$ corresponds to a higher probability (smaller $\delta$) that the average reward of the optimal action under $\pi$ exceeds that of the next-best action, indicating a more reliable decision. This aligns with the intuition that a larger $N$ corresponds to a closer approximation of the infinite-horizon MDP, where the random policy $\pi$ selects the same optimal action as that of the optimal policy (by Assumption 2). However, it is worth noting that Theorem 2 considers the worst-case guarantee of selecting the optimal actions over all states in the state space. It is therefore could be conservative. That being said, Theorem 2 still hints to the fact that one should select the action that yields the largest average reward in practice. In the practical implementation, we randomly select a query state from the state space and execute an episode of $N$ steps for each action in the action space. When the maximal return across all actions is no less than a pre-designed return threshold $\mathfrak{R}$, we choose the action that maximizes $\hat{Q}_{\mathrm{MDP}}^{\pi,N}(s_q, a)$ across the entire action space $A$. Otherwise, we randomly sample another query state and repeat the process of evaluating each action in a horizon $N$. The implementation details are summarized in Algorithm 3.

---

**Algorithm 3** Collecting Query States and Action Labels under Random Policy (MDP)

---

1: **Require:** Random policy $\pi$, state space $S$, action space $A$, environment $\tau$, trust horizon $N$, return threshold $\mathfrak{R}$
2: **Set** $\texttt{max\_return} = \mathfrak{R} - 1$
3: **while** $\texttt{max\_return} < \mathfrak{R}$ **do**
4:     Sample a query state $s_q \sim S$
5:     Empty a return list $L_r$
6:     **for** $a$ in $[A]$ **do**
7:         Initialize the state and action as $s_0 = s_q$, $a_0 = a$
8:         Run an episode of $N$ steps in $\tau$ under the random policy $\pi$
9:         Add the discounted episodic return to $L_r$
10:     **end for**
11:     $\texttt{max\_return} = \max(L_r)$
12: **end while**
13: Obtain $a_l = A(\arg\max(L_r))$
14: **Return** $(s_q, a_l)$

---

Notably, this work focuses on the grid world navigation problems where the reward is received solely upon reaching the unique goal. In this sparse reward MDP, it is worth highlighting that the action that maximizes the discounted return is essentially the same as that reaches the goal using fewest steps. This is because that the discounted return $\sum_t \gamma^t r_t$ has a term $\gamma^t$ where $t$ is the time step. Thus, fewer steps (smaller $t$) corresponds to larger discounted return (larger $\gamma^t$). Therefore, for any query state $s_q$, we prioritize actions that can achieve the goal within $N$ steps, with the actions consuming fewer steps being preferred. If no action can accomplish the goal within $N$ steps, we sample another query state until a qualified action is identified. This action is then designated as the action label associated with the query state $s_q$. Details of this implementation are outlined in Algorithm 5 (see Appendix B).

Having introduced the processes for collecting the context, query state, and action label under the random policy, we can now generate the pretraining dataset by integrating the aforementioned procedures (refer to Algorithm 4). Given the pretraining dataset, the model pretraining procedure as well as the offline and online deployment for SAD are summarized in Algorithm 6 (see Appendix B).

## 4.2 Performance Guarantees

This subsection establishes the performance guarantees of SAD, offering deeper insights into its efficacy.

**Corollary 1.** *Let hypotheses of Theorems 1 and 2 hold. Denote by $l$ the length of the trajectory. For any environment $\tau$ and history data $H$, SAD and the well-specified posterior sampling follow the same trajectory distribution with probability $(1 - \delta)^l$*

$$P_{\mathcal{F}_\theta}(trajectory \mid \tau, H) = P_{ps}(trajectory \mid \tau, H), \forall trajectory. \tag{13}$$

---

**Algorithm 4** State-Action Distillation (SAD) under Random Policy

---

1: **Require:** Empty pretraining dataset $\mathcal{D}$ with size $|\mathcal{D}|$, pretraining environment distribution $\mathcal{T}_{\text{train}}$, random policy $\pi$, context horizon length $T$, state space $S$, action space $A$, trust horizon $N$
2: **for** $i$ in $[|\mathcal{D}|]$ **do**
3:     Sample an environment $\tau \sim \mathcal{T}_{\text{train}}$
4:     Collect the context $\mathcal{C}$ under the environment $\tau$ and the random policy $\pi$ through Algorithm 1
5:     Collect the query state $s_q$ and the action label $a_l$ under the environment $\tau$ and the random policy $\pi$ through Algorithm 2 for MAB (Algorithm 3 for MDP)
6:     Add $(\mathcal{C}, s_q, a_l)$ to the pretraining dataset $\mathcal{D}$
7: **end for**
8: **Return** $\mathcal{D}$

---

*Proof.* See Appendix A.6. □

Notice that DPT takes the same distribution as that from a well-specified posterior sampling (Lee et al., 2024), which is widely recognized as a provably sample-efficient RL algorithm (Osband et al., 2013). However, it is crucial to highlight that our SAD approach focuses only on the pretraining dataset generation, while maintaining the same pretraining process as that of DPT. Building upon Theorems 1 and 2 implying that SAD probabilistically selects the optimal action (required by DPT), Corollary 1 substantiates that our SAD approach follows the same trajectory distribution as that of the posterior sampling with probability $(1-\delta)^l$.

Having established the corollary above, we next investigate the regret bound of SAD in the finite MDP setting (see details in Appendix A.7). Consider the online cumulative regret of SAD over $K$ episodes in the environment $\tau$ as $\text{Regret}_\tau(\mathcal{F}_\theta) = \sum_{k=0}^{K} V_\tau(\pi_\tau^*) - V_\tau(\pi_k)$, where $\pi_k(\cdot \mid s_t) = \mathcal{F}_\theta(\cdot \mid \mathcal{C}_{k-1}, s_t)$. Then, the regret bound of SAD is formally stated as follows.

**Corollary 2.** *Let hypotheses of Theorems 1 and 2 hold. Given environment $\tau$ and a constant $B' > 0$, suppose that $\sup_\tau \mathcal{T}_{test}(\tau)/\mathcal{T}_{train}(\tau) \leq B'$. In the finite MDP with horizon $T$, it holds with probability $(1-\delta)^{KT}$ that*

$$\mathbb{E}_{\mathcal{T}_{test}}[Regret_\tau(\mathcal{F}_\theta)] \leq \widetilde{\mathcal{O}}(B'|S|T^{3/2}\sqrt{K|A|}). \tag{14}$$

*Proof.* See Appendix A.8. □

Analogous to Corollary 1, Corollary 2 implies that SAD achieves the same regret bound as that of DPT (equation 14) with probability $(1-\delta)^{KT}$, as SAD and DPT undergo the identical pretraining process.

## 5 Experiments

In this section, we substantiate the efficacy of our proposed SAD method on five ICRL benchmark problems: *Gaussian Bandits, Bernoulli Bandits, Darkroom, Darkroom-Large, Miniworld*, which are commonly considered in the ICRL literature (Laskin et al., 2022; Lee et al., 2024; Dong et al., 2024). All these problems are challenging to solve in-context, as the test environments differ from the pretraining environments, while the parameters of the FM remain frozen during the test.

### 5.1 Environmental Setup

***Gaussian Bandits.*** We investigate a five-armed bandit problem in which the state space $S$ consists solely of a singleton state $s_q$. With each arm (action) pulled, the agent receives a reward. The goal is to identify the optimal arm that can maximize the cumulative reward. We consider the reward function for each arm following a Gaussian distribution with mean $\mu_a$ and variance $\sigma^2$, i.e., $R(\cdot|s_q, a) = \mathcal{N}(\mu_a, \sigma^2)$. Each arm possesses means $\mu_a$ drawn from a uniform distribution $U[0, 1]$ and all arms share the same variance $\sigma = 0.3$. We consider the pretraining and test data to have distinct Gaussian distributions with different means.

***Bernoulli Bandits.*** We adopt the same setup as in *Gaussian Bandits*, with the exception that the reward function does not follow a Gaussian distribution. Instead, we model the reward function using a Bernoulli

distribution. Specifically, the mean of each arm $\mu_a$ is drawn from a Beta distribution $Beta(1, 1)$, and the reward function follows a Bernoulli distribution with probability of success $\mu_a$. To validate the capability of SAD tackling OOD scenarios, we consider the test data drawn from the Bernoulli distribution while the pretraining data drawn from the Gaussian distribution as in the *Gaussian bandits*.

***Darkroom.*** *Darkroom* (Laskin et al., 2022; Zintgraf et al., 2019) is a two-dimensional navigation task with discrete state and action spaces. The room consists of $7 \times 7$ grids ($|S| = 49$), with an unknown goal randomly placed at any of these grids. The agent can select 5 actions: go up, go down, go left, go right, or stay. The horizon length for *Darkroom* is 49, meaning the agent must reach the goal within 49 moves. The challenge of this task arises from its sparse reward structure, i.e., the agent receives a reward of 1 solely upon reaching the unique goal, and 0 otherwise. Given $7 \times 7 = 49$ available goals, we utilize 39 of these goals ($\sim 80\%$) for pretraining and reserve the remaining 10 ($\sim 20\%$) (unseen during pretraining) for test.

***Darkroom-Large.*** We adopt the same setup as in *Darkroom*, yet with an expanded state space of $10 \times 10$ and a longer horizon $T = 100$. Consequently, the agent must explore the environment more extensively due to the sparse reward setting, making this task more challenging than *Darkroom*. We still consider $80\%$ of the 100 available goals for pretraining and the remaining unseen $20\%$ goals for test.

***Miniworld.*** *Miniworld* is a three-dimensional pixel-based navigation task. The agent is situated in a room with four differently colored boxes, one of which is the target (unknown to the agent). The agent must navigate to the target box using $25 \times 25 \times 3$ image observations and by selecting from 4 available actions: turn left, turn right, move forward, or stay. Similar to *Darkroom*, the agent receives a reward of 1 only upon approaching the unique target box, and 0 otherwise. The high-dimensional pixel inputs clearly render *Miniworld* a much more challenging task than *Darkroom* and *Darkroom-Large*.

## 5.2 Numerical Results

We consider four SOTA ICRL algorithms as our baselines in this work: AD, DIT, DPT, and DPT$^*$. Since all these methods are FM-based, we employ the same transformer architecture (causal GPT2 model (Radford et al., 2019)) and hyperparameters (number of attention layers, number of attention heads, embedding dimensions, etc) across all experiments to ensure a fair comparison. The main hyperparameters employed in this work are summarized in Tables 1-2 (refer to Appendix C.1).

In all experiments, we employ a uniform random policy to collect context, query states, and action labels, as indicated in Algorithms 1-3. Then, we pretrain the FM and deploy it with two options: online and offline. In online deployment, the pretrained model collects its own context by iteratively interacting with the test environments. In offline deployment, the pretrained model directly uses a sampled context from an offline context dataset, which is pre-collected using the random policy to interact with the test environments. The details of online/offline deployments are deferred to Algorithm 6 in Appendix B.

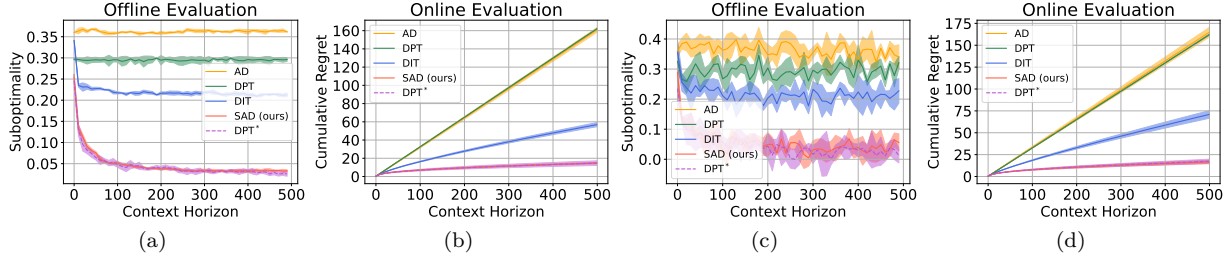

Figure 3: Offline and online evaluations of ICRL algorithms trained under a uniform random policy: AD, DPT, DIT, DPT$^*$, and SAD (ours). Each algorithm contains four independent runs with mean and standard deviation. *Gaussian Bandits*: (a) and (b), *Bernoulli Bandits*: (c) and (d).

**Bandits.** We adhere to offline and online evaluation metrics for *Bandits* established in (Lee et al., 2024). In the offline evaluation, we utilize the *suboptimality* over different context horizon, defined by $\mu_{a^*} - \mu_a$, where $\mu_{a^*}$ and $\mu_a$ represent the mean rewards over 200 test environments of the optimal arm and the selected arm, respectively. In online evaluation, we employ *cumulative regret*, defined by $\sum_{t=0}^{T}(\mu_{a^*} - \mu_{a_t})$, where $a_t$ denotes the selected arm at time $t$. Figures 3(a) and 3(b) demonstrate that our SAD approach significantly outperforms three SOTA baselines under uniform random policy, by achieving much lower *suboptimality* and *cumulative regret*. More specifically, let us define the performance improvement of SAD over baselines in the offline evaluation by $(suboptimality_{\text{baseline}} - suboptimality_{\text{SAD}})/suboptimality_{\text{SAD}}$. Likewise, the performance improvement in the online evaluation is to simply replace the *suboptimality* by *cumulative regret*. Then, SAD surpasses the best baseline, DIT, by achieving 354.0% performance improvements in the offline evaluation and 273.9% in the online evaluation (refer to the first row of Tables 3 and 4 in Appendix C.2). To evaluate the out-of-distribution performance of SAD compared to other baselines, we test the models pretrained by all methods on the *Gaussian Bandits* and assess their performance on the *Bernoulli Bandits* with no further fine-tuning. Figures 3(c) and 3(d) illustrate that SAD still achieves lower *suboptimality* and *cumulative regret* than all other baselines, demonstrating a more robust performance in handling out-of-distribution scenarios. More specifically, SAD surpasses the best baseline, DIT, by 289.5% in the offline evaluation and 313.9% in the online evaluation (refer to the second row of Tables 3 and 4 in Appendix C.2).

In the environments of *Darkroom* and *Miniworld*, we use return as the evaluation metric. Moreover, we define the performance improvement of our SAD approach over the baseline methods in both offline and online evaluations by $(Return_{\text{SAD}} - Return_{\text{baseline}})/Return_{\text{baseline}}$.

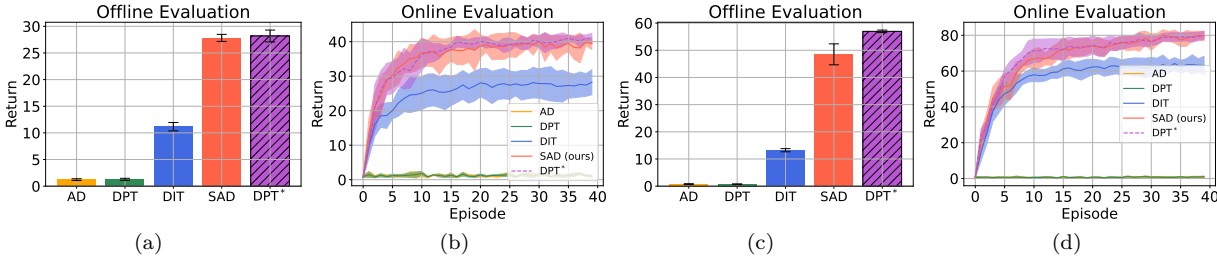

Figure 4: Offline and online evaluations of ICRL algorithms trained under a uniform random policy: AD, DPT, DIT, DPT*, and SAD (ours). Each algorithm contains four independent runs with mean and standard deviation. *DarkRoom*: (a) and (b). *DarkRoom-Large*: (c) and (d).

**Darkrooms.** Figure 4 demonstrates that our SAD approach significantly outperforms three SOTA baselines in the *Darkroom* and *Darkroom-Large* under uniform random policy, by achieving much higher return. In the *Darkroom*, SAD surpasses the best baseline, DIT, by 149.3% in the offline evaluation and 41.7% in the online evaluation (refer to the third row of Tables 3 and 4 in Appendix C.2). Likewise, in the *Darkroom-Large*, SAD outperforms the best baseline, DIT, by 266.8% in the offline evaluation and 24.7% in the online evaluation (refer to the fourth row of Tables 3 and 4 in Appendix C.2).

**Miniworld.** Figure 5 demonstrates that our SAD approach outperforms the three SOTA baselines in the *Miniworld* under uniform random policy, by achieving a higher return. More specifically, SAD surpasses the best baseline, DIT, by 122.1% in the offline evaluation and 21.7% in the online evaluation (refer to the fifth row of Tables 3 and 4 in Appendix C.2).

Tables 3 and 4 also imply that SAD significantly outperforms all baselines on average across the five ICRL benchmark environments. In the offline evaluation, SAD exceeds the best baseline DIT by 236.3% on average, the second-best DPT by 2015.9%, and the third-best AD by 2075.2%. In the online evaluation, SAD surpasses DIT by 135.2%, DPT by 3093.8%, and AD by 3208.8% on average. In addition to comparing SAD with the three SOTA ICRL algorithms under the uniform random policy, we also include the empirical performance of the DPT with optimal action labels (DPT*) as the oracle upper bound of SAD. We observe that SAD demonstrates performance comparable to DPT* in tasks such as *Gaussian Bandits*, *Bernoulli*

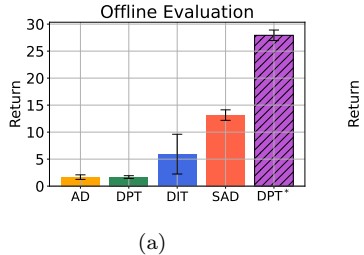
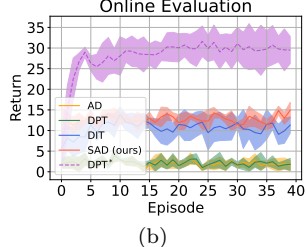

$$(a) \qquad\qquad (b)$$

Figure 5: Offline and online evaluations of ICRL algorithms trained under a uniform random policy: AD, DPT, DIT, DPT*, and SAD (ours). Each algorithm contains four independent runs with mean and standard deviation. Environment: *Miniworld*.

*Bandits*, *DarkRoom*, and *DarkRoom-Large*. Although *Miniworld* introduces challenges due to its pixel-based inputs and complex environments, SAD under the random policy still achieves approximately 50% of the performance of DPT*. Overall, SAD is within 18.6% of the performance of DPT* in the offline evaluation across five ICRL tasks, and within 12.3% in the online evaluation (see details in Tables 3 and 4).

## 5.3 Ablation Studies

**Trust Horizon.** Theorems 1 implies that the uniform random policy is probabilistically trustworthy within a horizon $N$, with monotonically increasing probability of selecting the optimal action with $N$ in the MAB problem. We substantiate this observation from the theorem by empirical evidence, as presented in Figure 6(a). Furthermore, we conduct empirical investigations into the influence of the trust horizon $N$ on the performance of the MAB problem, which considers the environments of *Gaussian Bandits*. As expected, a larger $N$ in the MAB problem leads to a better performance (see Figures 7(a) and 7(b)).

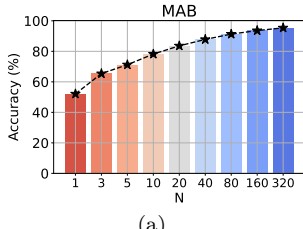
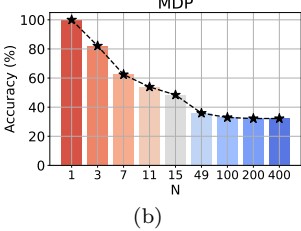
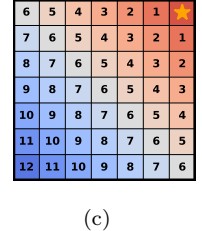
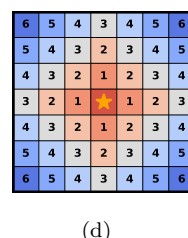

$$(a) \qquad\qquad (b) \qquad\qquad (c) \qquad\qquad (d)$$

Figure 6: (a) and (b): The accuracy (probability) of selecting the optimal action in the MAB and MDP problems with varying trust horizon $N$. (c) and (d): The minimal number of steps required for a query state to reach the goal (the golden star) in the upper-right corner and in the middle.

We then shift to the MDP problem with the environment of *Darkroom*. Notice that in our practical algorithm for the sparse-reward MDP like *Darkroom* (Algorithm 5), we only utilize the state-action pairs that can reach the goal within a trust horizon. Therefore, we solely record the probability/accuracy of selecting the optimal actions on those states, as presented in Figure 6(b). It shows that the accuracy monotonically decreases with respect to the trust horizon $N$, which, at first glance, may lead to monotonically decreasing performance as well. Nonetheless, we acknowledge that this is not the case. In particular, a large trust horizon $N$ in the MDP leads to the low accuracy of selecting the optimal action, whereas a small $N$ may induce the partially short-sighted training of FM, as Algorithm 5 solely trains on the states at most $N$ steps from the goal, instead of all states (refer to Figures 6(c) and 6(d)). Our numerical results in Figures 7(c) and 7(d) validate this with the fact $N = 7$ performing best, and provide the empirical evidence that either an excessively large or small trust horizon can lead to suboptimality.

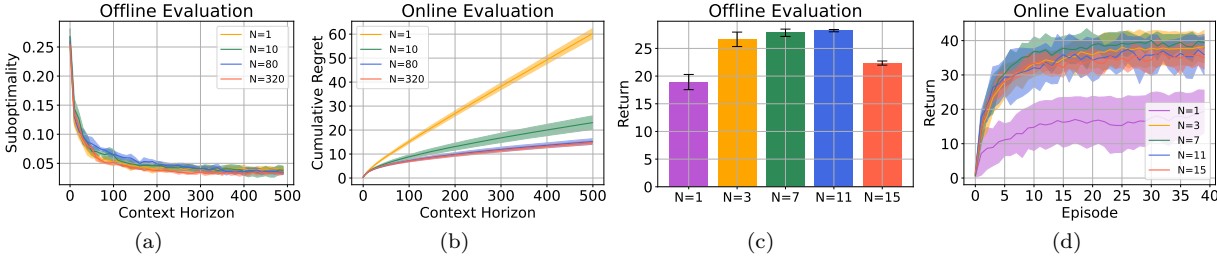

Figure 7: Offline and online evaluation of SAD with varying trust horizon $N$ for MAB: (a) and (b), and for MDP: (c) and (d). Each $N$ contains four independent runs with mean and standard deviation.

**Transformer Hyperparameters.** We aim to validate the robustness of our proposed SAD approach with respect to the hyperparameters in the transformer block. Concretely, we focus on the number of attention heads and attention layers, as they have large impacts on the model size of the transformer. As depicted in Figure 8, our empirical results in *Darkroom* demonstrate a robust performance across varying numbers of attention heads and attention layers.

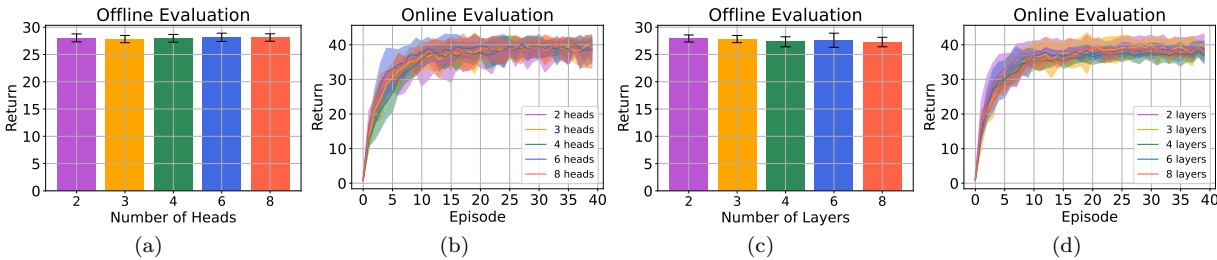

Figure 8: Offline and online evaluations of SAD with different transformer hyperparameters: the number of attention heads ((a) and (b)); the number of attention layers ((c) and (d)). Each hyperparameter contains four independent runs with mean and standard deviation.

# 6 Conclusion

In this work, we propose State-Action Distillation (SAD), a novel approach for generating the pretraining dataset for ICRL, which is designed to overcome the limitations imposed by the existing ICRL algorithms like AD, DPT, and DIT in terms of relying on well-trained or even optimal policies to collect the pretraining dataset. SAD leverages solely random policies to construct the pretraining data, significantly promoting the practical application of ICRL in real-world scenarios. We also provide the quantitative analysis of the trustworthiness as well as the performance guarantees of SAD. Moreover, our empirical results on multiple popular ICRL benchmark environments demonstrate significant improvements over the existing baselines in terms of both performance and robustness. Nevertheless, we note that SAD is currently limited to the discrete action space. Extending SAD to handle the continuous action space as well as more complex environments presents a promising direction for the future research.

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

# A Omitted Proofs

## A.1 Proof of Theorem 1

**Theorem 1** (MAB). *Let Assumption 1 hold. The random policy is at least $(1 - \delta)$-trustworthy as in Definition 1, when the trust horizon $N$ satisfies*

$$N \geq \frac{8B^2}{\left( \mathbb{E}[r(s_q, a^*)] - \max\limits_{a \in A \setminus \{a^*\}} \mathbb{E}[r(s_q, a)] \right)^2} \log \left( \frac{1 + \sqrt{1 - \delta}}{\delta} \right). \tag{15}$$

*Proof.* For any action $a \in A \setminus \{a^*\}$, consider two positive constants

$$\epsilon_1 = \alpha \left( \mathbb{E}[r(s_q, a^*)] - \mathbb{E}[r(s_q, a)] \right), \tag{16}$$

$$\epsilon_2 = (1 - \alpha) \left( \mathbb{E}[r(s_q, a^*)] - \mathbb{E}[r(s_q, a)] \right), \tag{17}$$

where $\alpha \in [0, 1]$.

Consider the following two inequalities

$$\frac{1}{N_{a^*}} \sum_{i=1}^{N_{a^*}} r(s_q, a^*) \geq \mathbb{E}[r(s_q, a^*)] - \epsilon_1, \tag{18}$$

$$\frac{1}{N_a} \sum_{i=1}^{N_a} r(s_q, a) \leq \mathbb{E}[r(s_q, a)] + \epsilon_2. \tag{19}$$

We note that the two inequalities above are the sufficient but not necessary conditions for $\frac{1}{N_{a^*}} \sum_{i=1}^{N_{a^*}} r(s_q, a^*) \geq \frac{1}{N_a} \sum_{i=1}^{N_a} r(s_q, a)$ to hold.

Therefore, we simply obtain that

$$P \left( \frac{1}{N_{a^*}} \sum_{i=1}^{N_{a^*}} r(s_q, a^*) \geq \frac{1}{N_a} \sum_{i=1}^{N_a} r(s_q, a) \right)$$

$$\geq P \left( \frac{1}{N_{a^*}} \sum_{i=1}^{N_{a^*}} r(s_q, a^*) \geq \mathbb{E}[r(s_q, a^*)] - \epsilon_1, \frac{1}{N_a} \sum_{i=1}^{N_a} r(s_q, a) \leq \mathbb{E}[r(s_q, a)] + \epsilon_2 \right) \tag{20}$$

$$= P \left( \frac{1}{N_{a^*}} \sum_{i=1}^{N_{a^*}} r(s_q, a^*) \geq \mathbb{E}[r(s_q, a^*)] - \epsilon_1 \right) \cdot P \left( \frac{1}{N_a} \sum_{i=1}^{N_a} r(s_q, a) \leq \mathbb{E}[r(s_q, a)] + \epsilon_2 \right), \tag{21}$$

where the last equation follows from the fact that each arm is independent to other arms. We then lower bound the two probabilities in the previous expression using Hoeffding's inequality (Hoeffding, 1994).

Since Assumption 1 implies that $r(\cdot, \cdot) \in [-B, B]$, Hoeffding's inequality yields

$$P \left( \frac{1}{N_a} \sum_{i=1}^{N_a} r(s_q, a) - \mathbb{E}[r(s_q, a)] \leq \epsilon_2 \right) \geq 1 - \exp \left( - \frac{2 N_a \epsilon_2^2}{(B - (-B))^2} \right), \tag{22}$$

i.e.,

$$P \left( \frac{1}{N_a} \sum_{i=1}^{N_a} r(s_q, a) - \mathbb{E}[r(s_q, a)] \leq \epsilon_2 \right) \geq 1 - \exp \left( - \frac{N_a \epsilon_2^2}{2 B^2} \right). \tag{23}$$

Likewise, we have

$$P \left( \frac{1}{N_{a^*}} \sum_{i=1}^{N_{a^*}} r(s_q, a^*) - \mathbb{E}[r(s_q, a^*)] \geq -\epsilon_1 \right) \geq 1 - \exp \left( - \frac{2 N_{a^*} \epsilon_1^2}{(B - (-B))^2} \right), \tag{24}$$

i.e.,

$$P\left(\frac{1}{N_{a^*}}\sum_{i=1}^{N_{a^*}} r(s_q, a^*) - \mathbb{E}[r(s_q, a^*)] \geq -\epsilon_1\right) \geq 1 - \exp\left(-\frac{N_{a^*}\epsilon_1^2}{2B^2}\right). \tag{25}$$

Since $N \triangleq \min_{a \in A} N_a$, it then follows from the monotonicity of the exponential function that

$$P\left(\frac{1}{N_{a^*}}\sum_{i=1}^{N_{a^*}} r(s_q, a^*) \geq \frac{1}{N_a}\sum_{i=1}^{N_a} r(s_q, a)\right)$$

$$\geq \left(1 - \exp\left(-\frac{N_{a^*}\epsilon_1^2}{2B^2}\right)\right) \cdot \left(1 - \exp\left(-\frac{N_a\epsilon_2^2}{2B^2}\right)\right) \tag{26}$$

$$\geq \left(1 - \exp\left(-\frac{N\epsilon_1^2}{2B^2}\right)\right) \cdot \left(1 - \exp\left(-\frac{N\epsilon_2^2}{2B^2}\right)\right). \tag{27}$$

Therefore, it holds for any $\alpha \in [0, 1]$ that

$$P\left(\frac{1}{N_{a^*}}\sum_{i=1}^{N_{a^*}} r(s_q, a^*) \geq \frac{1}{N_a}\sum_{i=1}^{N_a} r(s_q, a)\right)$$

$$\geq \left(1 - \exp\left(-\frac{N\alpha^2\left(\mathbb{E}[r(s_q, a^*)] - \mathbb{E}[r(s_q, a)]\right)^2}{2B^2}\right)\right) \cdot \left(1 - \exp\left(-\frac{N(1-\alpha)^2\left(\mathbb{E}[r(s_q, a^*)] - \mathbb{E}[r(s_q, a)]\right)^2}{2B^2}\right)\right).$$
$$\tag{28}$$

Notice that the maximal value of the previous equation with respect to $\alpha$ reaches at $\alpha = 0.5$. Then it holds that

$$P\left(\frac{1}{N_{a^*}}\sum_{i=1}^{N_{a^*}} r(s_q, a^*) \geq \frac{1}{N_a}\sum_{i=1}^{N_a} r(s_q, a)\right) \geq \left(1 - \exp\left(-\frac{N\left(\mathbb{E}[r(s_q, a^*)] - \mathbb{E}[r(s_q, a)]\right)^2}{8B^2}\right)\right)^2, \forall a \in A \setminus \{a^*\}. \tag{29}$$

Since the previous inequality holds for any $a \in A \setminus \{a^*\}$, let us define

$$\bar{a} = \arg\max_{a \in A \setminus \{a^*\}} \frac{1}{N_a}\sum_{i=1}^{N_a} r(s_q, a). \tag{30}$$

Then it also holds that

$$P\left(\frac{1}{N_{a^*}}\sum_{i=1}^{N_{a^*}} r(s_q, a^*) \geq \max_{a \in A \setminus \{a^*\}} \frac{1}{N_a}\sum_{i=1}^{N_a} r(s_q, a)\right)$$

$$\geq \left(1 - \exp\left(-\frac{N\left(\mathbb{E}[r(s_q, a^*)] - \mathbb{E}[r(s_q, \bar{a})]\right)^2}{8B^2}\right)\right)^2 \tag{31}$$

$$\geq \left(1 - \exp\left(-\frac{N\left(\mathbb{E}[r(s_q, a^*)] - \max_{a \in A \setminus \{a^*\}}\mathbb{E}[r(s_q, a)]\right)^2}{8B^2}\right)\right)^2. \tag{32}$$

where the last inequality follows from the monotonicity.

To make the previous probability greater than $1 - \delta$, we require

$$\left(1 - \exp\left(-\frac{N\left(\mathbb{E}[r(s_q, a^*)] - \max_{a \in A \setminus \{a^*\}}\mathbb{E}[r(s_q, a)]\right)^2}{8B^2}\right)\right)^2 \geq 1 - \delta. \tag{33}$$

Hence, we require the trust horizon $N \triangleq \min_{a \in A} N_a$ to satisfy

$$N \geq \frac{8B^2}{\left(\mathbb{E}[r(s_q, a^*)] - \max_{a \in A \setminus \{a^*\}} \mathbb{E}[r(s_q, a)]\right)^2} \log\left(\frac{1 + \sqrt{1 - \delta}}{\delta}\right). \tag{34}$$

This completes the proof.

$\square$

## A.2 Validity of Assumption 2 in the grid world MDP with a single sparse reward

For the sake of simplicity and without loss of generality, we consider a single-dimensional grid world MDP with the understanding that Assumption 2 holds for the two-dimensional grid world MDP as well, which is considered in our numerical experiments. The environmental details of the single-dimensional grid world MDP can be found in Figure 2, where the reward is received only upon reaching the unique goal. To proceed, we rely on the lemma below, and Assumption 2 is then validated by Proposition 1 that follows.

**Lemma 1.** *Consider the MDP of a single-dimensional grid world with $S = \{s_0, s_1, s_2, s_3, s_4\}$ and $A = \{a_0, a_1\}$, as depicted in Figure 2. Consider the random policy $\pi$ in Assumption 2. It holds that*

$$V_{MDP}^{\pi}(s_0) \geq V_{MDP}^{\pi}(s_1) \geq V_{MDP}^{\pi}(s_2) \geq V_{MDP}^{\pi}(s_3) \geq V_{MDP}^{\pi}(s_4). \tag{35}$$

*Proof.* For simplicity, we consider $\pi$ to be a uniform random policy, i.e., $P(a_0 \mid s) = P(a_1 \mid s) = 0.5, \forall s \in S$. Recall the Bellman expectation equation

$$V_{\mathrm{MDP}}^{\pi}(s) = \sum_{a \in A} \pi(a \mid s) \left(r(s, a) + \gamma \mathbb{E}_{s'} V_{\mathrm{MDP}}^{\pi}(s')\right). \tag{36}$$

By combining the previous Bellman expectation equation with Figure 2 yields

$$\begin{cases} V_{\mathrm{MDP}}^{\pi}(s_0) = \frac{1}{2}\left(r(s_0, a_0) + \gamma V_{\mathrm{MDP}}^{\pi}(s_0) + r(s_0, a_1) + \gamma V_{\mathrm{MDP}}^{\pi}(s_1)\right), \\ V_{\mathrm{MDP}}^{\pi}(s_1) = \frac{1}{2}\left(r(s_1, a_0) + \gamma V_{\mathrm{MDP}}^{\pi}(s_0) + r(s_1, a_1) + \gamma V_{\mathrm{MDP}}^{\pi}(s_2)\right), \\ V_{\mathrm{MDP}}^{\pi}(s_2) = \frac{1}{2}\left(r(s_2, a_0) + \gamma V_{\mathrm{MDP}}^{\pi}(s_1) + r(s_2, a_1) + \gamma V_{\mathrm{MDP}}^{\pi}(s_3)\right), \\ V_{\mathrm{MDP}}^{\pi}(s_3) = \frac{1}{2}\left(r(s_3, a_0) + \gamma V_{\mathrm{MDP}}^{\pi}(s_2) + r(s_3, a_1) + \gamma V_{\mathrm{MDP}}^{\pi}(s_4)\right), \\ V_{\mathrm{MDP}}^{\pi}(s_4) = \frac{1}{2}\left(r(s_4, a_0) + \gamma V_{\mathrm{MDP}}^{\pi}(s_3) + r(s_4, a_1) + \gamma V_{\mathrm{MDP}}^{\pi}(s_4)\right). \end{cases} \tag{37}$$

Substituting all rewards from Figure 9 (left) into the previous equations yields

$$\begin{cases} V_{\mathrm{MDP}}^{\pi}(s_0) = \frac{1}{2}\left(1 + \gamma V_{\mathrm{MDP}}^{\pi}(s_0) + \gamma V_{\mathrm{MDP}}^{\pi}(s_1)\right), \\ V_{\mathrm{MDP}}^{\pi}(s_1) = \frac{1}{2}\left(1 + \gamma V_{\mathrm{MDP}}^{\pi}(s_0) + \gamma V_{\mathrm{MDP}}^{\pi}(s_2)\right), \\ V_{\mathrm{MDP}}^{\pi}(s_2) = \frac{1}{2}\left(\gamma V_{\mathrm{MDP}}^{\pi}(s_1) + \gamma V_{\mathrm{MDP}}^{\pi}(s_3)\right), \\ V_{\mathrm{MDP}}^{\pi}(s_3) = \frac{1}{2}\left(\gamma V_{\mathrm{MDP}}^{\pi}(s_2) + \gamma V_{\mathrm{MDP}}^{\pi}(s_4)\right), \\ V_{\mathrm{MDP}}^{\pi}(s_4) = \frac{1}{2}\left(\gamma V_{\mathrm{MDP}}^{\pi}(s_3) + \gamma V_{\mathrm{MDP}}^{\pi}(s_4)\right). \end{cases} \tag{38}$$

Given $\gamma \in (0, 1)$, the last equation of equation 38 implies that

$$V_{\mathrm{MDP}}^{\pi}(s_3) = \frac{2 - \gamma}{\gamma} V_{\mathrm{MDP}}^{\pi}(s_4) \geq V_{\mathrm{MDP}}^{\pi}(s_4). \tag{39}$$

Then, the fourth equation of equation 38 yields

$$V_{\mathrm{MDP}}^{\pi}(s_2) = \frac{2}{\gamma} V_{\mathrm{MDP}}^{\pi}(s_3) - V_{\mathrm{MDP}}^{\pi}(s_4) \tag{40}$$

$$\geq \frac{2}{\gamma} V_{\mathrm{MDP}}^{\pi}(s_3) - V_{\mathrm{MDP}}^{\pi}(s_3) \tag{41}$$

$$\geq V_{\mathrm{MDP}}^{\pi}(s_3). \tag{42}$$

Likewise, the third equation of equation 38 can be rewritten as

$$V_{\text{MDP}}^{\pi}(s_1) = \frac{2}{\gamma} V_{\text{MDP}}^{\pi}(s_2) - V_{\text{MDP}}^{\pi}(s_3) \tag{43}$$

$$\geq \frac{2}{\gamma} V_{\text{MDP}}^{\pi}(s_2) - V_{\text{MDP}}^{\pi}(s_2) \tag{44}$$

$$\geq V_{\text{MDP}}^{\pi}(s_2). \tag{45}$$

Combining the previous inequality with the first two equations of equation 38 directly yields

$$V_{\text{MDP}}^{\pi}(s_0) \geq V_{\text{MDP}}^{\pi}(s_1). \tag{46}$$

This completes the proof.

$\square$

**Proposition 1.** *Consider the MDP of a single-dimensional grid world with $S = \{s_0, s_1, s_2, s_3, s_4\}$ and $A = \{a_0, a_1\}$, as depicted in Figure 2. Consider a uniform random policy $\pi$. It holds that*

$$\arg\max_{a \in A} Q_{MDP}^{\pi}(s, a) = \arg\max_{a \in A} Q_{MDP}^{*}(s, a), \, \forall s \in S. \tag{47}$$

*Proof.* By the definition of $Q_{\text{MDP}}^{*}$ and $\pi^*$ we obtain

$$Q_{\text{MDP}}^{*}(s, a) = Q_{\text{MDP}}^{\pi^*}(s, a), \, \forall (s, a) \in S \times A. \tag{48}$$

Since the objective of the agent in Figure 2 is to reach the goal state as quickly as possible, and stay still, $\pi^*(s)$ is given by

$$\pi^*(s) = a_0, \, \forall s \in S. \tag{49}$$

Moreover, we have

$$\arg\max_{a \in A} Q_{\text{MDP}}^{*}(s, a) = \arg\max_{a \in A} Q_{\text{MDP}}^{\pi^*}(s, a) = \pi^*(s) = a_0, \, \forall s \in S. \tag{50}$$

We then turn to consider the learning of Q-function under the random policy $\pi$. For simplicity, we consider $\pi$ to be a uniform random policy, i.e., $P(a_0 \mid s) = P(a_1 \mid s) = 0.5, \, \forall s \in S$.

We next prove that $Q_{\text{MDP}}^{\pi}(s, a_0) \geq Q_{\text{MDP}}^{\pi}(s, a_1), \, \forall s \in S$. We start with the state $s_0$. Notice that

$$Q_{\text{MDP}}^{\pi}(s_0, a_0) = r(s_0, a_0) + \gamma V_{\text{MDP}}^{\pi}(s_0) = 1 + \gamma V_{\text{MDP}}^{\pi}(s_0), \tag{51}$$

$$Q_{\text{MDP}}^{\pi}(s_0, a_1) = r(s_0, a_1) + \gamma V_{\text{MDP}}^{\pi}(s_1) = \gamma V_{\text{MDP}}^{\pi}(s_1). \tag{52}$$

Lemma 1 implies that $V_{\text{MDP}}^{\pi}(s_0) \geq V_{\text{MDP}}^{\pi}(s_1)$. The previous equations then directly indicate

$$Q_{\text{MDP}}^{\pi}(s_0, a_0) \geq Q_{\text{MDP}}^{\pi}(s_0, a_1). \tag{53}$$

For the state $s_1$, we have

$$Q_{\text{MDP}}^{\pi}(s_1, a_0) = r(s_1, a_0) + \gamma V_{\text{MDP}}^{\pi}(s_0) = 1 + \gamma V_{\text{MDP}}^{\pi}(s_0), \tag{54}$$

$$Q_{\text{MDP}}^{\pi}(s_1, a_1) = r(s_1, a_1) + \gamma V_{\text{MDP}}^{\pi}(s_2) = \gamma V_{\text{MDP}}^{\pi}(s_2). \tag{55}$$

Lemma 1 implies that $V_{\text{MDP}}^{\pi}(s_0) \geq V_{\text{MDP}}^{\pi}(s_1) \geq V_{\text{MDP}}^{\pi}(s_2)$. Then it holds that

$$Q_{\text{MDP}}^{\pi}(s_1, a_0) \geq Q_{\text{MDP}}^{\pi}(s_1, a_1). \tag{56}$$

For the state $s_2$, we have

$$Q_{\text{MDP}}^{\pi}(s_2, a_0) = r(s_2, a_0) + \gamma V_{\text{MDP}}^{\pi}(s_1) = \gamma V_{\text{MDP}}^{\pi}(s_1), \tag{57}$$

$$Q_{\text{MDP}}^{\pi}(s_2, a_1) = r(s_2, a_1) + \gamma V_{\text{MDP}}^{\pi}(s_3) = \gamma V_{\text{MDP}}^{\pi}(s_3). \tag{58}$$

Employing Lemma 1 directly yields

$$Q_{\text{MDP}}^{\pi}(s_2, a_0) \geq Q_{\text{MDP}}^{\pi}(s_2, a_1). \tag{59}$$

For the state $s_3$, we have

$$Q_{\text{MDP}}^{\pi}(s_3, a_0) = r(s_3, a_0) + \gamma V_{\text{MDP}}^{\pi}(s_2) = \gamma V_{\text{MDP}}^{\pi}(s_2), \tag{60}$$

$$Q_{\text{MDP}}^{\pi}(s_3, a_1) = r(s_3, a_1) + \gamma V_{\text{MDP}}^{\pi}(s_4) = \gamma V_{\text{MDP}}^{\pi}(s_4). \tag{61}$$

Employing Lemma 1 directly yields

$$Q_{\text{MDP}}^{\pi}(s_3, a_0) \geq Q_{\text{MDP}}^{\pi}(s_3, a_1). \tag{62}$$

Last but not least, for the state $s_4$, we have

$$Q_{\text{MDP}}^{\pi}(s_4, a_0) = r(s_4, a_0) + \gamma V_{\text{MDP}}^{\pi}(s_3) = \gamma V_{\text{MDP}}^{\pi}(s_3), \tag{63}$$

$$Q_{\text{MDP}}^{\pi}(s_4, a_1) = r(s_4, a_1) + \gamma V_{\text{MDP}}^{\pi}(s_4) = \gamma V_{\text{MDP}}^{\pi}(s_4). \tag{64}$$

Employing Lemma 1 directly yields

$$Q_{\text{MDP}}^{\pi}(s_4, a_0) \geq Q_{\text{MDP}}^{\pi}(s_4, a_1). \tag{65}$$

Hence we obtain

$$\arg\max_{a \in A} Q_{\text{MDP}}^{\pi}(s, a) = a_0, \ \forall s \in S. \tag{66}$$

and then

$$\arg\max_{a \in A} Q_{\text{MDP}}^{\pi}(s, a) = a_0 = \arg\max_{a \in A} Q_{\text{MDP}}^{*}(s, a), \ \forall s \in S. \tag{67}$$

This completes the proof.

$\square$

**Empirical validation of Assumption 2 in the grid world MDP.**  In addition to the theoretical proof above, we also provide an empirical validation of Assumption 2 in the grid world MDP in Figure 2.

We start by considering the Bellman expectation equation

$$Q_{\text{MDP}}^{\pi}(s, a) = r(s, a) + \gamma \mathbb{E}_{a'} \left[ Q_{\text{MDP}}^{\pi}(s', a') \right], \ \forall (s, a) \in S \times A. \tag{68}$$

Let us define the temporal-difference (TD) error as follows

$$\mathcal{E}(s, a) = \left| r(s, a) + \gamma \mathbb{E}_{a'} \left[ Q_{\text{MDP}}^{\pi}(s', a') \right] - Q_{\text{MDP}}^{\pi}(s, a) \right|, \ \forall (s, a) \in S \times A. \tag{69}$$

With the understanding that the TD error $\mathcal{E}(s, a)$ of $Q_{\text{MDP}}^{\pi}(s, a)$ is zero, we iteratively learn and update $Q_{\text{MDP}}^{\pi}(s, a)$ by minimizing the TD error. To that end, consider $\gamma = 0.99$ and a convergence threshold $\epsilon_Q = 10^{-6}$ (can be arbitrarily small). We initialize $Q_{\text{MDP}}^{\pi}(s, a)$ to be full of zeros as in Figure 9 (middle-left). Subsequently, we consistently update $Q_{\text{MDP}}^{\pi}(s, a)$ under the uniform policy $\pi$, until convergence as follows

$$\mathcal{E}(s, a) \leq \epsilon_Q, \ \forall (s, a) \in S \times A. \tag{70}$$

Empirically, we observe that the Q-table $Q_{\text{MDP}}^{\pi}(s, a)$ converges after the $1216^{th}$ iteration; demonstrated in Figure 10. As depicted in Figure 9 (middle-right), the convergent $Q_{\text{MDP}}^{\pi}(s, a)$ implies that the optimal action for any state under the uniform policy $\pi$ is $a_0$ (see the golden stars), i.e.,

$$\arg\max_{a \in A} Q_{\text{MDP}}^{\pi}(s, a) = a_0, \ \forall s \in S. \tag{71}$$

Likewise, we follow the same process above using the Bellman optimality equation

$$Q^*_{\text{MDP}}(s,a) = r(s,a) + \gamma \max_{a' \in A} Q^*_{\text{MDP}}(s',a'), \ \forall (s,a) \in S \times A. \tag{72}$$

The convergent $Q^*_{\text{MDP}}(s,a)$ in Figure 9 (right) demonstrates that the optimal action for any state under the optimal policy is $a_0$ as well, i.e.,

$$\arg\max_{a \in A} Q^*_{\text{MDP}}(s,a) = a_0, \ \forall s \in S. \tag{73}$$

This validates Assumption 2 empirically.

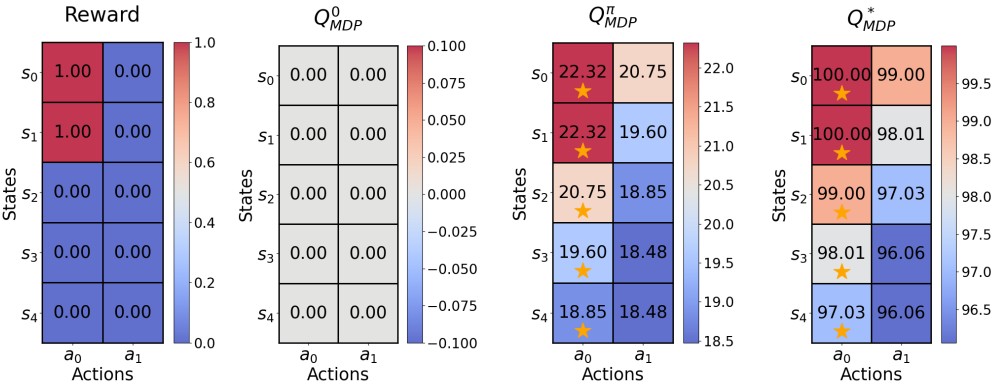

Figure 9: Left: reward table. Middle-Left: initial Q-table. Middle-Right: convergent Q-table under uniform policy $\pi$. Right: convergent optimal Q-table.

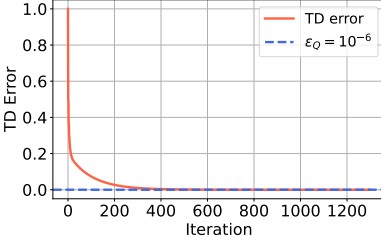

Figure 10: The learning curve of the maximal TD error of $Q^\pi_{\text{MDP}}(s,a)$ over the entire state-action spaces: $\max_{(s,a) \in S \times A} |r(s,a) + \gamma \mathbb{E}_{a'} [Q^\pi_{\text{MDP}}(s',a')] - Q^\pi_{\text{MDP}}(s,a)|$.

## A.3  Discussion of Assumption 2 in the grid world MDP with two sparse rewards

As depicted in Figure 11, we consider the same problem setting as in Appendix A.2 except that we assign the rewards $R_l$ and $R_s$ to the states $s_0$ and $s_4$, respectively. In the following three cases, we demonstrate that Assumption 2 holds for some (but not all) navigation problems that have two sparse rewards. The validity of Assumption 2 depends on the problem-specific factors such as the rewards and the discount factor $\gamma$.

(a) Consider $R_l = 1, R_s = 0.1, \gamma = 0.99$. The action that maximizes the Q table for both random and optimal policies is the same, i.e., $a_0 = \arg\max_a Q^\pi_{MDP}(s,a) = \arg\max_a Q^*_{MDP}(s,a)$ for all states as shown in Figure 12.

(b) Consider $R_l = 1, R_s = 0.9, \gamma = 0.8$. In this case, the random policy selects $a_0$ at the states $s_0, s_1, s_2$ and $a_1$ at the states $s_3, s_4$ (see $Q^\pi_{MDP}$ in Figure 13). However, notice that the optimal policy selects exactly the same optimal actions as that of the random policy, i.e., $a_0$ for states $s_0, s_1, s_2$ and $a_1$ for states $s_3, s_4$ (see $Q^*_{MDP}$ in Figure 13), thus Assumption 2 still holding.

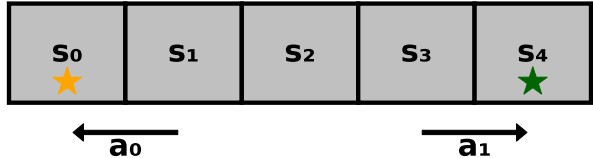

Figure 11: A single-dimensional grid world MDP comprising five states $\{s_0, s_1, s_2, s_3, s_4\}$. The environment offers two possible actions: $a_0$ that corresponds to moving left, and $a_1$ that corresponds to moving right. Crossing the boundaries is strictly prohibited. Any transitions that would result in boundary crossing will be confined to the current position. The reward structure is sparse, with a value of $R_l$ received upon reaching $s_0$, a value of $R_s$ received upon reaching $s_4$ and a value of 0 otherwise. We consider an infinite time horizon with a discounter factor $\gamma$.

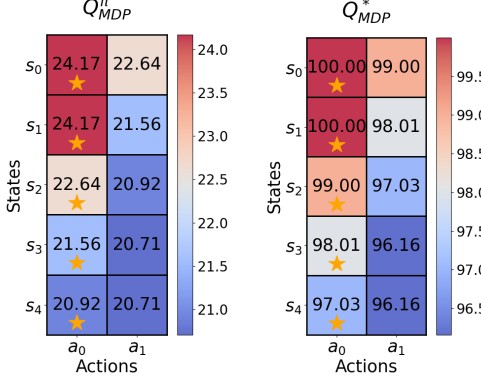

Figure 12: When $R_l = 1, R_s = 0.1, \gamma = 0.99$. Left: convergent Q-table under uniform policy $\pi$. Right: convergent optimal Q-table.

(c) Consider $R_l = 1, R_s = 0.9, \gamma = 0.97$. The convergent Q-tables in Figure 14 indicate that under the random policy, the optimal action for states $s_3$ and $s_4$ is $a_1$, whereas under the optimal policy, it is $a_0$. This discrepancy indicates that Assumption 2 does not hold.

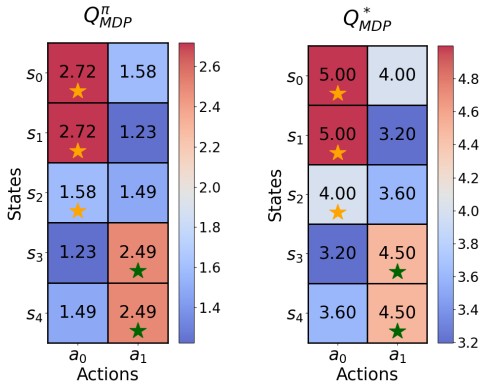

Figure 13: When $R_l = 1, R_s = 0.9, \gamma = 0.8$. Left: convergent Q-table under uniform policy $\pi$. Right: convergent optimal Q-table.

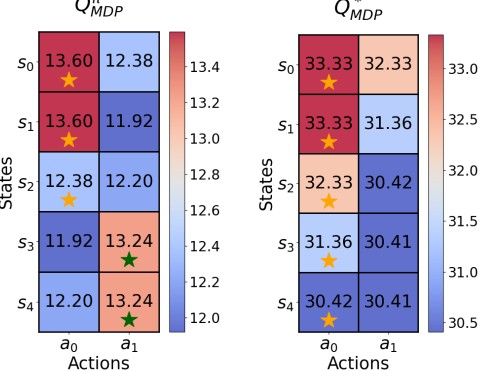

Figure 14: When $R_l = 1, R_s = 0.9, \gamma = 0.97$. Left: convergent Q-table under uniform policy $\pi$. Right: convergent optimal Q-table.

### A.4 Proof of Theorem 2

**Theorem 2** (MDP). *Let Assumptions 1 and 2 hold. Define $\kappa = \min_{s_q \in S} \left( Q_{MDP}^{\pi}(s_q, a^*) - \max_{a \in A \setminus \{a^*\}} Q_{MDP}^{\pi}(s_q, a) \right)$. Consider the trust horizon $N > \log_\gamma \left( \kappa(1-\gamma)/(2B) \right) - 1$. The random policy is at least $(1-\delta)$-trustworthy as in Definition 2, when the number of episodes $N_{ep}$ satisfies*

$$N_{ep} \geq \underbrace{\frac{2\left(1 - \gamma^{N+1}\right)^2}{\left(\kappa\left(1-\gamma\right)/(2B) - \gamma^{N+1}\right)^2}}_{G_1} \log\left(\frac{1 + \sqrt{1-\delta}}{\delta}\right). \tag{74}$$

*Proof.* For any action $a \in A \setminus \{a^*\}$, let us define

$$Q_{\mathrm{MDP}}^{\pi}(s_q, a) = \underbrace{\mathbb{E}\left[\sum_{t=0}^{N} \gamma^t r(s_t, a_t) \mid s_0 = s_q, a_0 = a\right]}_{Q_{\mathrm{MDP}}^{\pi,N}(s_q,a)} + \underbrace{\mathbb{E}\left[\sum_{t=N+1}^{\infty} \gamma^t r(s_t, a_t) \mid s_0 = s_q, a_0 = a\right]}_{\xi_a}, \tag{75}$$

$$Q_{\mathrm{MDP}}^{\pi}(s_q, a^*) = \underbrace{\mathbb{E}\left[\sum_{t=0}^{N} \gamma^t r(s_t, a_t) \mid s_0 = s_q, a_0 = a^*\right]}_{Q_{\mathrm{MDP}}^{\pi,N}(s_q,a^*)} + \underbrace{\mathbb{E}\left[\sum_{t=N+1}^{\infty} \gamma^t r(s_t, a_t) \mid s_0 = s_q, a_0 = a^*\right]}_{\xi_{a^*}}. \tag{76}$$

Then $\forall s_q \in S$,

$$Q_{\mathrm{MDP}}^{\pi,N}(s_q, a^*) - Q_{\mathrm{MDP}}^{\pi,N}(s_q, a) \tag{77}$$

$$= Q_{\mathrm{MDP}}^{\pi}(s_q, a^*) - \xi_{a^*} - (Q_{\mathrm{MDP}}^{\pi}(s_q, a) - \xi_a) \tag{78}$$

$$= Q_{\mathrm{MDP}}^{\pi}(s_q, a^*) - Q_{\mathrm{MDP}}^{\pi}(s_q, a) + \xi_a - \xi_{a^*} \tag{79}$$

$$= Q_{\mathrm{MDP}}^{\pi}(s_q, a^*) - Q_{\mathrm{MDP}}^{\pi}(s_q, a) + \sum_{t=N+1}^{\infty} \gamma^t \left( \mathbb{E}\left[r(s_t, a_t) \mid s_0 = s_q, a_0 = a\right] - \mathbb{E}\left[r(s_t, a_t) \mid s_0 = s_q, a_0 = a^*\right] \right) \tag{80}$$

$$\overset{(a)}{\geq} Q_{\mathrm{MDP}}^{\pi}(s_q, a^*) - Q_{\mathrm{MDP}}^{\pi}(s_q, a) - \gamma^{N+1} \frac{2B}{1-\gamma} \tag{81}$$

$$\geq Q_{\mathrm{MDP}}^{\pi}(s_q, a^*) - \max_{a \in A \setminus \{a^*\}} Q_{\mathrm{MDP}}^{\pi}(s_q, a) - \gamma^{N+1} \frac{2B}{1-\gamma} \tag{82}$$

$$\geq \min_{s_q \in S} \left( Q_{\mathrm{MDP}}^{\pi}(s_q, a^*) - \max_{a \in A \setminus \{a^*\}} Q_{\mathrm{MDP}}^{\pi}(s_q, a) \right) - \gamma^{N+1} \frac{2B}{1-\gamma} \tag{83}$$

$$\overset{(b)}{=} \kappa - \gamma^{N+1} \frac{2B}{1-\gamma}, \tag{84}$$

$$\overset{(c)}{>} 0. \tag{85}$$

where $(a)$ follows from Assumption 1 and the properties of geometry series, $(b)$ follows from the definition of $\kappa$, $(c)$ is due to $N > \log_\gamma \left( \kappa(1-\gamma)/(2B) \right) - 1$. Therefore, let us consider two positive constants as follows

$$\epsilon_1 = \alpha \left( Q_{\mathrm{MDP}}^{\pi,N}(s_q, a^*) - Q_{\mathrm{MDP}}^{\pi,N}(s_q, a) \right), \tag{86}$$

$$\epsilon_2 = (1-\alpha) \left( Q_{\mathrm{MDP}}^{\pi,N}(s_q, a^*) - Q_{\mathrm{MDP}}^{\pi,N}(s_q, a) \right), \tag{87}$$

where $\alpha \in [0, 1]$. Consider the following two inequalities

$$\hat{Q}_{\mathrm{MDP}}^{\pi,N}(s_q, a^*) \geq Q_{\mathrm{MDP}}^{\pi,N}(s_q, a^*) - \epsilon_1, \tag{88}$$

$$\hat{Q}_{\mathrm{MDP}}^{\pi,N}(s_q, a) \leq Q_{\mathrm{MDP}}^{\pi,N}(s_q, a) + \epsilon_2. \tag{89}$$

We acknowledge that equation 88 and equation 89 are the sufficient but not necessary conditions for $\hat{Q}^{\pi,N}_{\mathrm{MDP}}(s_q, a^*) \geq \hat{Q}^{\pi,N}_{\mathrm{MDP}}(s_q, a)$ to hold. Thus,

$$P\left(\hat{Q}^{\pi,N}_{\mathrm{MDP}}(s_q, a^*) \geq \hat{Q}^{\pi,N}_{\mathrm{MDP}}(s_q, a)\right) \tag{90}$$

$$\geq P\left(\hat{Q}^{\pi,N}_{\mathrm{MDP}}(s_q, a^*) \geq Q^{\pi,N}_{\mathrm{MDP}}(s_q, a^*) - \epsilon_1, \hat{Q}^{\pi,N}_{\mathrm{MDP}}(s_q, a) \leq Q^{\pi,N}_{\mathrm{MDP}}(s_q, a) + \epsilon_2\right) \tag{91}$$

$$= P\left(\hat{Q}^{\pi,N}_{\mathrm{MDP}}(s_q, a^*) \geq Q^{\pi,N}_{\mathrm{MDP}}(s_q, a^*) - \epsilon_1\right) \cdot P\left(\hat{Q}^{\pi,N}_{\mathrm{MDP}}(s_q, a) \leq Q^{\pi,N}_{\mathrm{MDP}}(s_q, a) + \epsilon_2\right) \tag{92}$$

where the last equation follows from the fact that each action is independent to other actions.

Assumption 1 implies that $\sum_{t=0}^{N}\left(\gamma^t r(s_t, a_t) \mid s_0 = s_q, a_0 = a, \pi\right) \in [-B\frac{1-\gamma^{N+1}}{1-\gamma}, B\frac{1-\gamma^{N+1}}{1-\gamma}], \forall a \in A$. We therefore lower bound the two probabilities in the previous expression using Hoeffding's inequality (Hoeffding, 1994)

$$P\left(\hat{Q}^{\pi,N}_{\mathrm{MDP}}(s_q, a^*) - Q^{\pi,N}_{\mathrm{MDP}}(s_q, a^*) \geq -\epsilon_1\right) \geq 1 - \exp\left(-\frac{N_{\mathrm{ep}}\epsilon_1^2}{2B^2\left(\frac{1-\gamma^{N+1}}{1-\gamma}\right)^2}\right), \tag{93}$$

$$P\left(\hat{Q}^{\pi,N}_{\mathrm{MDP}}(s_q, a) - Q^{\pi,N}_{\mathrm{MDP}}(s_q, a) \leq \epsilon_2\right) \geq 1 - \exp\left(-\frac{N_{\mathrm{ep}}\epsilon_2^2}{2B^2\left(\frac{1-\gamma^{N+1}}{1-\gamma}\right)^2}\right). \tag{94}$$

Then, it holds for any $\alpha \in [0, 1]$ that

$$P\left(\hat{Q}^{\pi,N}_{\mathrm{MDP}}(s_q, a^*) \geq \hat{Q}^{\pi,N}_{\mathrm{MDP}}(s_q, a)\right) \tag{95}$$

$$\geq \left(1 - \exp\left(-\frac{N_{\mathrm{ep}}\epsilon_1^2}{2B^2\left(\frac{1-\gamma^{N+1}}{1-\gamma}\right)^2}\right)\right) \cdot \left(1 - \exp\left(-\frac{N_{\mathrm{ep}}\epsilon_2^2}{2B^2\left(\frac{1-\gamma^{N+1}}{1-\gamma}\right)^2}\right)\right) \tag{96}$$

$$\geq \left(1 - \exp\left(-\frac{N_{\mathrm{ep}}\alpha^2\left(Q^{\pi,N}_{\mathrm{MDP}}(s_q, a^*) - Q^{\pi,N}_{\mathrm{MDP}}(s_q, a)\right)^2}{2B^2\left(\frac{1-\gamma^{N+1}}{1-\gamma}\right)^2}\right)\right) \tag{97}$$

$$\cdot \left(1 - \exp\left(-\frac{N_{\mathrm{ep}}(1-\alpha)^2\left(Q^{\pi,N}_{\mathrm{MDP}}(s_q, a^*) - Q^{\pi,N}_{\mathrm{MDP}}(s_q, a)\right)^2}{2B^2\left(\frac{1-\gamma^{N+1}}{1-\gamma}\right)^2}\right)\right) \tag{98}$$

Notice that the maximal value of the previous expression with respect to $\alpha$ reaches at $\alpha = 0.5$. Then it holds that

$$P\left(\hat{Q}^{\pi,N}_{\mathrm{MDP}}(s_q, a^*) \geq \hat{Q}^{\pi,N}_{\mathrm{MDP}}(s_q, a)\right) \geq \left(1 - \exp\left(-\frac{N_{\mathrm{ep}}\left(Q^{\pi,N}_{\mathrm{MDP}}(s_q, a^*) - Q^{\pi,N}_{\mathrm{MDP}}(s_q, a)\right)^2}{8B^2\left(\frac{1-\gamma^{N+1}}{1-\gamma}\right)^2}\right)\right)^2, \forall a \in A \setminus \{a^*\}. \tag{99}$$

Since the previous inequality holds for any $a \in A \setminus \{a^*\}$, let us define

$$\hat{a} = \underset{a \in A \setminus \{a^*\}}{\arg\max} \ \hat{Q}^{\pi,N}_{\mathrm{MDP}}(s_q, a). \tag{100}$$

Then it also holds that

$$P\left(\hat{Q}_{\text{MDP}}^{\pi,N}(s_q, a^*) \geq \max_{a \in A \setminus \{a^*\}} \hat{Q}_{\text{MDP}}^{\pi,N}(s_q, a)\right)$$

$$\geq \left(1 - \exp\left(-\frac{N_{\text{ep}}\left(Q_{\text{MDP}}^{\pi,N}(s_q, a^*) - Q_{\text{MDP}}^{\pi,N}(s_q, \hat{a})\right)^2}{8B^2\left(\frac{1-\gamma^{N+1}}{1-\gamma}\right)^2}\right)\right)^2 \tag{101}$$

$$\geq \left(1 - \exp\left(-\frac{N_{\text{ep}}\left(Q_{\text{MDP}}^{\pi,N}(s_q, a^*) - \max_{a \in A \setminus \{a^*\}} Q_{\text{MDP}}^{\pi,N}(s_q, a)\right)^2}{8B^2\left(\frac{1-\gamma^{N+1}}{1-\gamma}\right)^2}\right)\right)^2, \tag{102}$$

where the last inequality follows from the monotonicity and the fact that $Q_{\text{MDP}}^{\pi,N}(s_q, a^*) - Q_{\text{MDP}}^{\pi,N}(s_q, a) \geq 0, \forall a \in A \setminus \{a^*\}$.

Let us define

$$\bar{a} = \arg\max_{a \in A \setminus \{a^*\}} Q_{\text{MDP}}^{\pi,N}(s_q, a). \tag{103}$$

Then $\forall s_q \in S$ we obtain

$$Q_{\text{MDP}}^{\pi,N}(s_q, a^*) - \max_{a \in A \setminus \{a^*\}} Q_{\text{MDP}}^{\pi,N}(s_q, a) \tag{104}$$

$$\overset{(a)}{=} Q_{\text{MDP}}^{\pi,N}(s_q, a^*) - Q_{\text{MDP}}^{\pi,N}(s_q, \bar{a}) \tag{105}$$

$$\overset{(b)}{=} Q_{\text{MDP}}^{\pi}(s_q, a^*) - \xi_{a^*} - (Q_{\text{MDP}}^{\pi}(s_q, \bar{a}) - \xi_{\bar{a}}) \tag{106}$$

$$= Q_{\text{MDP}}^{\pi}(s_q, a^*) - Q_{\text{MDP}}^{\pi}(s_q, \bar{a}) + \xi_{\bar{a}} - \xi_{a^*} \tag{107}$$

$$= Q_{\text{MDP}}^{\pi}(s_q, a^*) - Q_{\text{MDP}}^{\pi}(s_q, \bar{a}) + \sum_{t=N+1}^{\infty} \gamma^t \left(\mathbb{E}\left[r(s_t, a_t) \mid s_0 = s_q, a_0 = \bar{a}\right] - \mathbb{E}\left[r(s_t, a_t) \mid s_0 = s_q, a_0 = a^*\right]\right) \tag{108}$$

$$\overset{(c)}{\geq} Q_{\text{MDP}}^{\pi}(s_q, a^*) - Q_{\text{MDP}}^{\pi}(s_q, \bar{a}) - \gamma^{N+1}\frac{2B}{1-\gamma} \tag{109}$$

$$\overset{(d)}{\geq} Q_{\text{MDP}}^{\pi}(s_q, a^*) - \max_{a \in A \setminus \{a^*\}} Q_{\text{MDP}}^{\pi}(s_q, a) - \gamma^{N+1}\frac{2B}{1-\gamma} \tag{110}$$

$$\geq \min_{s_q \in S}\left(Q_{\text{MDP}}^{\pi}(s_q, a^*) - \max_{a \in A \setminus \{a^*\}} Q_{\text{MDP}}^{\pi}(s_q, a)\right) - \gamma^{N+1}\frac{2B}{1-\gamma} \tag{111}$$

$$\overset{(e)}{=} \kappa - \gamma^{N+1}\frac{2B}{1-\gamma} \tag{112}$$

$$\overset{(f)}{>} 0, \tag{113}$$

where $(a)$ follows from the definition of $\bar{a}$, $(b)$ follows from the definition of $Q_{\text{MDP}}^{\pi}$, $(c)$ follows from Assumption 1 and the properties of geometry series, $(d)$ follows from the fact that $\bar{a}$ may not be the maximizer of $Q_{\text{MDP}}^{\pi}(s_q, \cdot)$, $(e)$ follows from the definition of $\kappa$, $(f)$ is due to $N > \log_\gamma\left(\kappa(1-\gamma)/(2B)\right) - 1$.

Consequently, equation 102 is monotonically increasing with $Q_{\text{MDP}}^{\pi,N}(s_q, a^*) - \max_{a \in A \setminus \{a^*\}} Q_{\text{MDP}}^{\pi,N}(s_q, a)$. Substituting the previous inequality into equation 102 and by the monotonicity yields

$$P\left(\hat{Q}_{\text{MDP}}^{\pi,N}(s_q, a^*) \geq \max_{a \in A \setminus \{a^*\}} \hat{Q}_{\text{MDP}}^{\pi,N}(s_q, a)\right) \geq \left(1 - \exp\left(-\frac{N_{\text{ep}}\left(\kappa - \gamma^{N+1}\frac{2B}{1-\gamma}\right)^2}{8B^2\left(\frac{1-\gamma^{N+1}}{1-\gamma}\right)^2}\right)\right)^2, \tag{114}$$

To make the previous probability greater than $1 - \delta$, we require

$$\left( 1 - \exp\left( -\frac{N_{\mathrm{ep}} \left( \kappa - \gamma^{N+1} \frac{2B}{1-\gamma} \right)^2}{8B^2 \left( \frac{1-\gamma^{N+1}}{1-\gamma} \right)^2} \right) \right)^2 \geq 1 - \delta. \tag{115}$$

Thus,

$$N_{\mathrm{ep}} \geq \log\left( \frac{1 + \sqrt{1-\delta}}{\delta} \right) \frac{8B^2 \left( \frac{1-\gamma^{N+1}}{1-\gamma} \right)^2}{\left( \kappa - \gamma^{N+1} \frac{2B}{1-\gamma} \right)^2} \tag{116}$$

$$= \frac{2 \left( 1 - \gamma^{N+1} \right)^2}{\left( \kappa \left( 1 - \gamma \right) / (2B) - \gamma^{N+1} \right)^2} \log\left( \frac{1 + \sqrt{1-\delta}}{\delta} \right). \tag{117}$$

This completes the proof.

$\square$

## A.5 Technical Lemma

**Lemma 2.** *Given $N > \log_\gamma \left( \kappa(1-\gamma)/(2B) \right) - 1$, $G_1$ in equation 12 is monotonically decreasing with respect to the trust horizon $N$.*

*Proof.* We proceed by defining $Y = \kappa(1-\gamma)/(2B)$. Since $N > \log_\gamma \left( \kappa(1-\gamma)/(2B) \right) - 1$, we can obtain

$$Y = \frac{\kappa \left( 1 - \gamma \right)}{2B} \in (\gamma^{N+1}, 1]. \tag{118}$$

Let $Z = N + 1$, and then we can rewrite $G_1$ as

$$G_1 = \frac{2 \left( 1 - \gamma^Z \right)^2}{(Y - \gamma^Z)^2}, \text{ where } Y \in (\gamma^Z, 1]. \tag{119}$$

The chain rule implies that

$$\frac{\partial G_1}{\partial N} = \frac{\partial G_1}{\partial Z} \cdot \frac{\partial Z}{\partial N} \tag{120}$$

$$= \frac{\partial G_1}{\partial Z} \tag{121}$$

$$= 4 \left( \frac{1 - \gamma^Z}{Y - \gamma^Z} \right) \frac{-\gamma^Z \log \gamma (Y - \gamma^Z) + (1 - \gamma^Z)\gamma^Z \log \gamma}{(Y - \gamma^Z)^2} \tag{122}$$

$$= 4 \left( \frac{1 - \gamma^Z}{Y - \gamma^Z} \right) \frac{\gamma^Z \log \gamma (1 - Y)}{(Y - \gamma^Z)^2} \tag{123}$$

$$\leq 0, \tag{124}$$

where the last inequality follows from $\gamma \in (0, 1]$ and $Y \in (\gamma^Z, 1]$.

This completes the proof.

$\square$

## A.6 Proof of Corollary 1

**Corollary 1.** *Let hypotheses of Theorems 1 and 2 hold. Denote by $l$ the length of the trajectory. For any environment $\tau$ and history data $H$, SAD and the well-specified posterior sampling follow the same trajectory*

distribution with probability $(1 - \delta)^l$

$$P_{\mathcal{F}_\theta}(\text{trajectory} \mid \tau, H) = P_{ps}(\text{trajectory} \mid \tau, H), \forall \text{trajectory}. \tag{125}$$

*Proof.* To proceed, we rely on the following assumption.

**Assumption 3.** *Denote by $\mathcal{F}_\theta$ the pretrained FM. $\forall(\mathcal{C}, s_q)$, assume $P_{train}(a \mid \mathcal{C}, s_q) = \mathcal{F}_\theta(a \mid \mathcal{C}, s_q)$ for all $a \in A$.*

Note that Assumption 3 is a common assumption in the in-context learning literature (Xie et al., 2021; Lee et al., 2024), assuming that the pretrained FM fits the pretraining distribution exactly provided with sufficient coverage and data, where the SAD fits with a sufficiently large trust horizon $N$.

With Assumption 3 established, Theorem 1 of (Lee et al., 2024) implies that equation 125 holds when the optimal action is selected at each step. In addition, Theorems 1 and 2 indicate that the FM trained by SAD selects the optimal action label with probability $1 - \delta$ at each step. Consequently, equation 125 holds for SAD with probability $(1 - \delta)^l$.

This completes the proof. $\square$

### A.7 Finite MDP setting from Osband et al. (2013)

Let us consider the finite MDP setting as in (Osband et al., 2013), where $\mathbb{E}[r(s_t, a_t)] \in [0, 1]$. Denote by $S, A, T$ the state space, action space, and time horizon. Consider the uniform random policy $\pi$ for sampling the context $\mathcal{C}$ and query state $s_q$. Denote by $\mathcal{T}_{\text{test}}(\tau)$ and $\mathcal{T}_{\text{train}}(\tau)$ the test and pretraining distribution over the environment $\tau$, respectively. Consider the online cumulative regret of SAD over $K$ episodes in the environment $\tau$ as

$$\text{Regret}_\tau(\mathcal{F}_\theta) \stackrel{\Delta}{=} \sum_{k=0}^{K} V_\tau(\pi_\tau^*) - V_\tau(\pi_k), \tag{126}$$

where $\pi_k(\cdot \mid s_t) = \mathcal{F}_\theta(\cdot \mid \mathcal{C}_{k-1}, s_t)$.

### A.8 Proof of Corollary 2

**Corollary 2**. *Let hypotheses of Theorems 1 and 2 hold. Given the environment $\tau$ and a constant $B' > 0$, suppose that $\sup_\tau \mathcal{T}_{test}(\tau)/\mathcal{T}_{train}(\tau) \leq B'$. In the finite MDP setting above, it holds with probability $(1-\delta)^{KT}$ that*

$$\mathbb{E}_{\mathcal{T}_{test}}[Regret_\tau(\mathcal{F}_\theta)] \leq \widetilde{\mathcal{O}}(B'|S|T^{3/2}\sqrt{K|A|}). \tag{127}$$

*Proof.* Theorems 1 and 2 imply that the FM trained by SAD selects the optimal action label with probability $1 - \delta$ at each step, while the finite MDP setting above comprises $K \cdot T$ steps. Therefore, it holds with probability $(1 - \delta)^{KT}$ that the trained FM $\mathcal{F}_\theta$ is equivalent to the posterior sampling established in Corollary 1. Then, it follows directly from Corollary 6.2 of (Lee et al., 2024) that with probability $(1 - \delta)^{KT}$ it holds that

$$\mathbb{E}_{\mathcal{T}_{\text{train}}}[\text{Regret}_\tau(\mathcal{F}_\theta)] \leq \widetilde{\mathcal{O}}(|S|T^{3/2}\sqrt{K|A|}), \tag{128}$$

where the notation $\widetilde{\mathcal{O}}$ omits the polylogarithmic dependence. Subsequently, by using the bounded likelihood ratio between the test and pretraining distributions yields

$$\mathbb{E}_{\mathcal{T}_{\text{test}}}[\text{Regret}_\tau(\mathcal{F}_\theta)] = \int \mathcal{T}_{\text{test}}(\tau)\text{Regret}_\tau(\mathcal{F}_\theta)d(\tau) \tag{129}$$

$$\leq B' \int \mathcal{T}_{\text{train}}(\tau)\text{Regret}_\tau(\mathcal{F}_\theta)d(\tau) \tag{130}$$

$$= B'\mathbb{E}_{\mathcal{T}_{\text{train}}}[\text{Regret}_\tau(\mathcal{F}_\theta)] \tag{131}$$

$$\leq \widetilde{\mathcal{O}}(B'|S|T^{3/2}\sqrt{K|A|}). \tag{132}$$

This completes the proof. $\square$

## B    Implementation Details

We provide below the pseudo-codes that are omitted in the main body of the paper.

---

**Algorithm 5** Collecting Query States and Action Labels under Random Policy (Sparse-Reward MDP)

---

1: **Require:** Random policy $\pi$, state space $S$, action space $A$, environment $\tau$, trust horizon $N$
2: **Set** min_step $= N + 1$
3: **while** min_step $> N$ **do**
4:     Sample a query state $s_q \sim S$
5:     Empty a step list $L_s$
6:     **for** $a$ in $[A]$ **do**
7:         Initialize the state and action as $s_0 = s_q$, $a_0 = a$
8:         Run an episode of $N$ steps in $\tau$ under the random policy $\pi$, and terminate the episode early upon receiving a reward
9:         Add consumed steps to $L_s$ (add "$N + 1$" if no reward is received)
10:     **end for**
11:     min_step $= \min(L_s)$
12: **end while**
13: Obtain $a_l = A(\arg\min(L_s))$
14: **Return** $(s_q, a_l)$

---

**Algorithm 6** Pretraining and Deployment of SAD (Inspired by (Lee et al., 2024))

---

1: **Require:** Pretraining dataset $\mathcal{D}$, initial model parameters $\theta$, test environment distribution $\mathcal{T}_{\text{test}}$, number of episodes $N_E$
2: **// Model pretraining**
3: **while** not converged **do**
4:     Sample $(\mathcal{C}, s_q, a_l)$ from the pretraining dataset $\mathcal{D}$ and predict actions by the model $\mathcal{F}_\theta(\cdot|\mathcal{C}_i, s_q)$ for all $i \in [|\mathcal{C}|]$
5:     Compute the loss in equation 3 with respect to the action label $a_l$ and backpropagate to update $\theta$.
6: **end while**
7: **// Offline deployment**
8: Sample unseen environments $\tau \sim \mathcal{T}_{\text{test}}$
9: Sample a context $\mathcal{C} \sim \mathcal{T}_{\text{test}}(\cdot \mid \tau)$
10: Deploy $\mathcal{F}_\theta$ in $\tau$ by selecting $a_t \in \arg\max_{a \in A} \mathcal{F}_\theta(a \mid \mathcal{C}, s_t)$ at time step $t$
11: **// Online deployment**
12: Sample unseen environments $\tau \sim \mathcal{T}_{\text{test}}$ and initialize empty context $\mathcal{C} = \{\}$
13: **for** $i$ in $[N_E]$ **do**
14:     Deploy $\mathcal{F}_\theta$ by sampling $a_t \sim \mathcal{F}_\theta(\cdot \mid \mathcal{C}, s_t)$ at time step $t$
15:     Add $(s_0, a_0, r_0, \ldots)$ to $\mathcal{C}$
16: **end for**

---

## C    Experimental Details

### C.1    Hyperparameters

The main hyperparameters employed in this work are summarized in Tables 1-2.

### C.2    Additional Results

We provide in this subsection additional experimental results that are omitted in the main body of paper.

Table 1: The main hyperparameters of each algorithm

| Hyperparameters | AD | DPT | DIT | SAD (ours) |
|---|---|---|---|---|
| Causal transformer | GPT2 | GPT2 | GPT2 | GPT2 |
| Number of attention heads | 3 | 3 | 3 | 3 |
| Number of attention layers | 3 | 3 | 3 | 3 |
| Embedding size | 32 | 32 | 32 | 32 |
| DIT Weight $\lambda$ | N/A | N/A | 500 | N/A |
| Learning rate | 0.001 | 0.001 | 0.001 | 0.001 |
| Dropout | 0.1 | 0.1 | 0.1 | 0.1 |

Table 2: The main hyperparameters of each environment

| Hyperparameters | *Gaussian Bandits* | *Bernoulli Bandits* | *Darkroom* | *Darkroom-Large* | *Miniworld* |
|---|---|---|---|---|---|
| Action dimension | 5 | 5 | 5 | 5 | 4 |
| Pixel-based | ✗ | ✗ | ✗ | ✗ | ✓ |
| Trust Horizon | 320 | 320 | 7 | 10 | 3 |
| # of epochs | 100 | 100 | 100 | 100 | 200 |
| Context horizon | 500 | 500 | 49 | 100 | 50 |
| Training/Test ratio | 0.8/0.2 | 0.8/0.2 | 0.8/0.2 | 0.8/0.2 | 0.8/0.2 |
| # of environments | 100000 | 100000 | 24010 | 100000 | 40000 |

Figure 15 presents the dataset generation time of our SAD approach, compared with other SOTA baselines (AD, DPT, DIT), where the MAB and MDP problems consider the environments of *Gaussian Bandits* and *Darkroom* respectively. SAD (avg) represents the average of consumed time over different trust horizons (consistent with those in Figure 6). On average, SAD requires the most significant amount of time in the MDP problem. While in the MAB problem, SAD ranks as the second most time-consuming method, with its duration surpassing all except DIT. Notice that the additional computational time of SAD aligns with the prevailing trend of leveraging increased computation to fully harness the advanced reasoning capabilities of FMs (Brown, 2020).

Figure 16 compares our SAD approach with the SOTA baselines (AD, DPT, DIT) under the same time budget (38 seconds, which is consumed by DIT). Despite with degraded performance and increased vairance compared to before, our SAD method still outperforms all baselines, especially in the offline deployment. This implies that our method has a more robust performance than the baselines when tackling the random contexts as the offline deployment uses pre-collected random contexts.

Table 5 demonstrates that our method with $N = 7$ uses 21,445,147 transition data only and consumes 82 seconds only in the dataset generation. In contrast, obtaining well-trained policies for all pretraining environments in the baseline methods (AD) uses 2,401,000,000 transition data and consumes 3452 seconds in the dataset generation. Furthermore, the baseline methods need to learn and maintain policy and value networks for all pretraining environments (additional costs), which is not the case in our method since the only thing we do is collecting at random. It is true that the baseline methods take about 20-40 seconds (see Figure 15) if they just employ the random policy (no well-trained policies), however, accompanied by a catastrophic performance as depicted in Figure 4.

Table 3: Performance improvements of SAD compared to baseline algorithms in the offline evaluation.

| Environment | SAD vs AD | SAD vs DPT | SAD vs DIT | SAD vs DPT* |
|---|---|---|---|---|
| *Gaussian Bandits* | 647.0% | 508.9% | 354.0% | −5.2% |
| *Bernoulli Bandits* | 553.4% | 426.8% | 289.5% | −18.7% |
| *Darkroom* | 2162.2% | 2069.6% | 149.3% | −1.3% |
| *Darkroom-Large* | 6325.5% | 6389.9% | 266.8% | −14.9% |
| *Miniworld* | 687.7% | 684.2% | 122.1% | −52.9% |
| Average | 2075.2% | 2015.9% | 236.3% | −18.6% |

Table 4: Performance improvements of SAD compared to baseline algorithms in the online evaluation.

| Environment | SAD vs AD | SAD vs DPT | SAD vs DIT | SAD vs DPT* |
|---|---|---|---|---|
| *Gaussian Bandits* | 933.6% | 942.5% | 273.9% | −0.4% |
| *Bernoulli Bandits* | 846.8% | 830.9% | 313.9% | −0.2% |
| *Darkroom* | 3053.9% | 2893.8% | 41.7% | −3.4% |
| *Darkroom-Large* | 10626.9% | 10221.8% | 24.7% | −0.1% |
| *Miniworld* | 582.9% | 580.1% | 21.7% | −57.3% |
| Average | 3208.8% | 3093.8% | 135.2% | −12.3% |

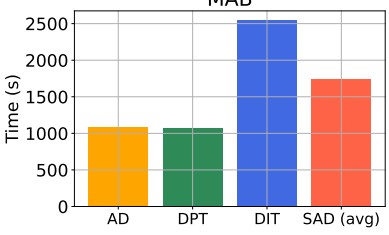 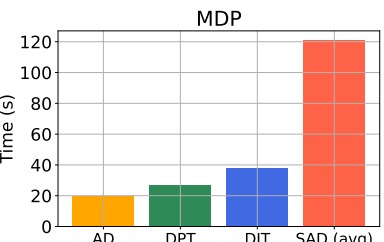

Figure 15: The dataset generation time consumed by SAD averaged over varying trust horizons (as in Figure 6), compared with AD, DPT, and DIT.

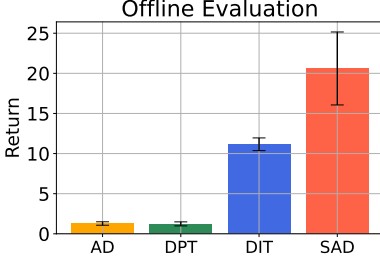 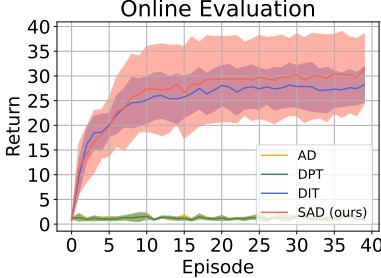

Figure 16: Offline and online evaluations of ICRL algorithms under the same time budget (38 seconds).

Table 5: The amount of transition data and computation time of SAD ($N = 7$) compared to AD with well-trained policies.

| Algorithms | Amount of Transition Data | Computation Time ($s$) |
|---|---|---|
| AD | $2,401,000,000$ | 3452 |
| SAD (ours) | $21,445,147$ | 82 |

