# OpenReview forum: "Random Policy Enables In-Context Reinforcement Learning within Trust Horizons"
_TMLR — Accepted by TMLR_

### Review · Reviewer_T9U3 · 2025-02-06

**Summary Of Contributions:**

This work focuses on in-context reinforcement learning (ICRL), where previous methods require a well-trained or optimal policy during the pretraining phase. It proposes a novel approach, State-Action Distillation (SAD), which uses a random policy to explore good state-action pairs via a trust horizon. The effectiveness of SAD is demonstrated from both theoretical and empirical perspectives.

**Audience:**

Yes

**Broader Impact Concerns:**

No ethical concerns are noted.

**Claims And Evidence:**

Yes

**Requested Changes:**

Figure 1 depicts the overall framework of the proposed SAD. However, it does not clearly demonstrate how SAD works, especially the concept of the Trust Horizon, which is the core of SAD but is not adequately explained in the current figure. A clearer illustration describing the algorithm flow is encouraged.

**Strengths And Weaknesses:**

**Strengths:**
1. This work addresses the important issue of previous ICRL methods requiring an optimal policy by using a random policy to obtain action labels.
2. It provides theoretical support for performance guarantees and evaluates the proposed method through comprehensive experiments, achieving remarkable improvements over baselines.

**Weaknesses:**
1. To obtain action labels, SAD needs to explore the entire action space within the N trust horizon. This process appears computationally expensive and may lead to unsafe situations in certain environments.
2. This work aims to improve ICRL using foundational models. However, the experiments are conducted only with GPT-2, a relatively small-scale model. The performance of SAD with larger models, such as 7B or larger, remains unexplored.

---

> ### Author Response · Authors · 2025-02-10
> **Response to Reviewer T9U3**
>
> We thank the reviewer for providing insightful and constructive reviews. We are glad to hear that in the reviewer’s opinion our work addresses important issues in the ICRL about requiring well-trained or optimal policies for pretraining, accompanied with theoretical supports and comprehensive experimental validations.
>
> The concerns and suggestions raised by the reviewer can significantly improve the quality of our paper. Detailed responses to the specific questions raised by the reviewer follow.
>
> 1. ''*To obtain action labels, SAD needs to explore the entire action space within the N trust horizon. This process appears computationally expensive and may lead to unsafe situations in certain environments.*''
> - **Response:** We thank the reviewer for pointing this out. We would like to address the reviewer's concern regarding the computation cost and unsafe scenarios separately.
> **(i)** **Compared with AD/DPT/DIT**, we indeed discuss and compare in Appendix D.3 the computation cost of our SAD approach over other SOTA ICRL baselines in terms of the computational time. On average, we found that SAD requires the most significant amount of time in the MDP problem. While in the MAB problem, SAD ranks as the second most time-consuming method, with its duration surpassing all except DIT. In fact, given the superior performance of SAD over the baselines, the additional computation cost of SAD can be employed while  retaining overall computational efficiencies. **Compared with DPT$^\star$**, Figures 2 and 3 show that SAD achieves the comparable performance with that of DPT$^\star$, which, however, demands optimal action labels. As we emphasize throughout the paper, obtaining such optimal action labels for a substantial volume of pretraining environments could be prohibitively expensive, as it necessitates substantial training for a single optimal action label.  Since our SAD method samples once per action only, it trades off the computation cost in a more efficient way than DPT$^\star$.
> **(ii)** In unconstrained RL, unsafe decisions/actions are common issues since RL algorithms like Q-learning requires to explore the entire state-action space infinite times to guarantee the convergence. To this end, safe/constrained RL aims to impose safety constraints to the unconstrained RL such that the agent can avoid unsafe and risky decisions. While we think ICRL with safety constraints is an interesting direction for future research, it falls beyond the scope of this work.
>
> 2. ''*This work aims to improve ICRL using foundational models. However, the experiments are conducted only with GPT-2, a relatively small-scale model. The performance of SAD with larger models, such as 7B or larger, remains unexplored.*''
> - **Response:** We sincerely appreciate the reviewer’s comments. We would like to clarify this from the following three aspects:
> **(i)** In the literature of ICRL and transformer-based RL, it is very common to consider GPT-2 as the backbone model, e.g., in AD, DPT, DIT, DT (Decision Transformer), ODT (Online Decision Transformer) and Prompt-DT (Prompt Decision Transformer). Therefore, we also consider GPT-2 model to ensure a fair comparison with our ICRL baselines (AD, DPT, DIT) which use GPT-2.
> **(ii)** Due to our computational resource limitations, we are not able to run 7B or larger LLMs. However, we agree with the reviewer that implementing SAD with larger model is a promising direction for future research.
> **(iii)** We **do agree** with the reviewer that it is significant to study how the model size would affect SAD's performance. In Figure 8, we study this by focusing on the number of attention heads and layers, as they have large impacts on the model size of Transformers. We investigated 2 heads, 3 heads, 4 heads, 2 layers, 3 layers and 4 layers. After the reviewer's comment, we are currently studying the effect of **more layers and more heads**, and we have finished preliminary experiments of **6 heads, 8 heads** and **6 layers, 8 layers**. We will include these results in the revised version of the paper. These preliminary results demonstrate that our SAD has a robust performance with respect to the model size, which aligns with the findings presented in AD and DPT papers.
>
> 3. ''*Figure 1 depicts the overall framework of the proposed SAD. However, it does not clearly demonstrate how SAD works, especially the concept of the Trust Horizon, which is the core of SAD but is not adequately explained in the current figure. A clearer illustration describing the algorithm flow is encouraged.*''
> - **Response:** Thank you for bringing this to our attention. We will include a more detailed schematic in the revised version of our paper and will effectively demonstrate how SAD works by explaining the Trust Horizon clearer.
>
> We are grateful for the reviewers' thoughtful feedback, which will guide us in strengthening the manuscript. The revisions will enhance the rigor, clarity, and contextualization of our contributions.

---

> > ### Comment · Reviewer_T9U3 · 2025-03-12
> > **Comment by Reviewer T9U3**
> >
> > Thank authors for their thoughtful response. I have no further questions at this time.

---

> > > ### Author Response · Authors · 2025-03-12
> > > **Response to Reviewer T9U3**
> > >
> > > You are very welcome. We sincerely appreciate the reviewer’s comments and time invested in the review, and we hope this discussion will lead to meaningful contributions to the community.

---

> > > > ### Author Response · Authors · 2025-03-21
> > > > **Thank you note**
> > > >
> > > > Dear Reviewer T9U3,
> > > >
> > > > As we approach the end of the rebuttal period, we would like to express our sincere gratitude once again for your insightful comments and constructive feedback. **We hope that our responses and the revised manuscript have addressed your concerns**. If you have any additional comments, please feel free to let us know.

---

### Review · Reviewer_4kRM · 2025-02-16

**Summary Of Contributions:**

1. The paper presents a method to train foundation models (FMs) for in-context reinforcement learning (ICRL) using random contexts and random query states.
2. In addition to constructing the context and query state, the paper presents a method to obtain action labels. It proposes three distinct procedures:
   1. For the multi-armed bandit (MAB) case, choose the action that yields the maximum average reward after each action has been taken N times.
   2. For MDPs with sparse rewards, select the action that requires the fewest steps to reach a goal. Restart the search if no action can achieve the goal in fewer than N steps.
   3. For MDPs with dense rewards, sample trajectories of length N—referred to as the ‘trust horizon’.

**Audience:**

Yes

**Broader Impact Concerns:**

I do not have any concerns.

**Claims And Evidence:**

Yes

**Requested Changes:**

1. Please reorganize the paper so that the critical components are included in the main text rather than being relegated to the appendix (see the comments above for specific examples).
2. Please introduce and discuss the RL environments in the main paper. Additionally, explain why these environments were chosen (i.e., what makes them particularly interesting and challenging).
3. Please provide a more detailed discussion of Assumption 2 in the main text, addressing the concerns noted above.
4. Please include a discussion of Corollaries 1 and 2. How do they compare with results in the existing literature? The discussion should help the reader contextualize whether these results are truly useful.
5. You justified not using AD by noting that it only allows short episode lengths. However, why can’t a longer context length be used? For the domains under consideration, how long would the context need to be, and why would that be prohibitive?
6. I am curious why samples beyond the N collected for computing the average reward (in Algorithm 2, step 2) are discarded. Why would extra samples be problematic? Moreover, if exactly N samples per action are needed, why is there a need to sample from a random policy at all? Wouldn’t a simple for-loop suffice?
7. For Theorem 1, the notation is inconsistent: you sum over the index $i \in {1, \ldots, N}$, yet there is no corresponding index in the reward function. Additionally, the reward function is denoted as $r(s,a)$ in one instance and as $r(s,a \mid \pi)$ in another. Please ensure that the notation is consistent.
8. Please provide explicit definitions for ‘trust horizon’ and ‘trustworthiness’ to ensure clarity for the reader.

**Strengths And Weaknesses:**

Strengths:

1. The paper is conceptually sound.
2. It also addresses an important aspect of ICRL by relaxing the context and policy requirements.
3. The results appear promising in the presented domains.


Weaknesses:
1. The paper seems artificially constrained to 12 pages, resulting in a non-linear and confusing presentation. For example, the related work—as well as some referenced algorithms and results—has been relegated to the appendix. Moreover, the motivating example for the main assumption (Assumption 2) appears only later in the paper.
2. Assumption 2 is very strong. While I understand it holding in the example provided, and in certain navigation tasks and gridworlds, introducing it six pages into the paper is problematic. Because it is critical to the mathematical framework, it should be discussed in greater detail in the main text, and the scope of the applicable domains should be outlined in the introduction. I am curious what the authors think about this, and if I am misunderstanding something.
3. The definition of ‘trust horizon’ is confusing due to its different interpretations in the contexts of MABs and MDPs.
4. The setup for sparse rewards is not clearly defined. It is implicitly assumed to be the case in which a reward is received only upon reaching the goal. However, what about cases where a reward is obtained only twice in a trajectory? Would that not also constitute a sparse reward setting? And which action-labeling algorithm should practitioners use in such cases?
5. The paper does not discuss Corollaries 1 and 2.

6. The motivating example involving healthcare applications in the introduction appears misplaced. **How is a uniform policy a reasonable choice in healthcare?** Moreover, how does the healthcare example relate to Assumption 2, and can it satisfy that assumption? The authors should either revise the motivating example or ensure consistency by justifying Assumption 2 as well as the use of a uniform random policy in healthcare applications.

---

> ### Author Response · Authors · 2025-02-21
> **Response to Reviewer 4kRM (Part 1)**
>
> We thank the reviewer for providing a thoughtful review with detailed feedback. We are glad to learn that in the reviewer’s opinion our work is conceptually sound, addresses a critical challenge in ICRL (in terms of the context and policy requirements), and exhibits promising results.
>
> In addition, we believe the concerns and requested changes raised by the reviewer can significantly enhance our paper. Detailed responses to each of the comments/requested changes raised by the reviewer follow.
>
> 1. ''*The paper seems artificially constrained to 12 pages, resulting in a non-linear and confusing presentation. For example, the related work—as well as some referenced algorithms and results—has been relegated to the appendix. Moreover, the motivating example for the main assumption (Assumption 2) appears only later in the paper.''
> --> ''Please reorganize the paper so that the critical components are included in the main text rather than being relegated to the appendix (see the comments above for specific examples).*''
>
> - **Response:** We thank the reviewer for bringing up this issue, and we agree that the critical components mentioned by the reviewer are significant and can enhance the clarity of our presentation. We will **reorganize** the paper to include those critical components in the main texts in our revised version of paper.
>
>
> 2. ''*--> Please introduce and discuss the RL environments in the main paper. Additionally, explain why these environments were chosen (i.e., what makes them particularly interesting and challenging).*''
>
> - **Response:** We thank the reviewer for pointing this out. In the revised version of our paper, we will incorporate a detailed introduction and discussion of the RL environments considered in this work within the main text. The reason to select these environments is **twofold**. First, they serve as the **benchmark environments** since they have been previously considered in the ICRL literature (e.g., [1, 2, 3]). Equally important is that these environments are **challenging to solve in-context**. In particular, the test environments differ from the pretraining environments, and the foundation model (FM) parameters **remain frozen** during the test. The challenge is even more pronounced in the Darkroom and MiniWorld environments due to their **sparse-reward** structure (where the rewards are received only upon achieving the goal) and **high-dimensional pixel inputs**.
>
>   [1] Lee, J., Xie, A., Pacchiano, A., Chandak, Y., Finn, C., Nachum, O., & Brunskill, E. (2024). Supervised pretraining can learn in-context reinforcement learning. Advances in Neural Information Processing Systems, 36.
>
>   [2] Dong, J., Guo, M., Fang, E. X., Yang, Z., & Tarokh, V. In-Context Reinforcement Learning Without Optimal Action Labels. In ICML 2024 Workshop on In-Context Learning.
>
>   [3] Laskin, M., Wang, L., Oh, J., Parisotto, E., Spencer, S., Steigerwald, R., ... & Mnih, V. (2022). In-context reinforcement learning with algorithm distillation. arXiv preprint arXiv:2210.14215.
>
> 3. ''*Assumption 2 is very strong. While I understand it holding in the example provided, and in certain navigation tasks and gridworlds, introducing it six pages into the paper is problematic. Because it is critical to the mathematical framework, it should be discussed in greater detail in the main text, and the scope of the applicable domains should be outlined in the introduction. I am curious what the authors think about this, and if I am misunderstanding something.
> --> Please provide a more detailed discussion of Assumption 2 in the main text, addressing the concerns noted above.*''
>
> - **Response:** This is a very good point raised by the reviewer, and we agree that it is crucial to incorporate more detailed discussion of Assumption 2 in the main texts. Moreover, we agree that outlining the applicable domains (especially in the Introduction) will more clearly demonstrate the specific areas of our focus in this work. In addition, and related to the first requested modification, incorporating the navigation motivating example in the main body of the paper will naturally bring the exposition of Assumption 2 **earlier** in the new version of the manuscript.

---

> > ### Author Response · Authors · 2025-02-21
> > **Response to Reviewer 4kRM (Part 2)**
> >
> > 4. ''*The paper does not discuss Corollaries 1 and 2.
> > --> Please include a discussion of Corollaries 1 and 2. How do they compare with results in the existing literature? The discussion should help the reader contextualize whether these results are truly useful.*''
> > - **Response:** We thank the reviewer for bringing this to our attention. We agree that comparing with the results in the literature of state-of-the-art algorithms is important and helpful for readers. As part of the reorganization effort of the paper discussed in the first point of this response, we will incorporate further discussion on these results. In particular, our method focuses only on the pretraining dataset generation, while maintaining the same pretraining process as that of DPT[1]. Note that DPT requires the optimal action labels in the pretraining dataset and our Theorems 1 and 2 in the manuscript guarantee that our method selects the optimal action labels with **high probability**. Accordingly, our method achieves the same performance guarantees as that of DPT with **high probability**.
> >
> >   [1] Lee, J., Xie, A., Pacchiano, A., Chandak, Y., Finn, C., Nachum, O., & Brunskill, E. (2024). Supervised pretraining can learn in-context reinforcement learning. Advances in Neural Information Processing Systems, 36.
> >
> >
> > 5. ''*You justified not using AD by noting that it only allows short episode lengths. However, why can’t a longer context length be used? For the domains under consideration, how long would the context need to be, and why would that be prohibitive?*''
> >
> > - **Response:** The original AD paper[1] claims that AD is designed to learn the improvement operator of the source RL algorithm, thus necessitating that the context captures across-episodic information. However, given the **limited context length** of common transformer models, tasks with **very long** episodes are not well-suited for AD, as the transformer cannot capture across-episodic information in such scenarios. We will explain this more explicitly in the revised manuscript.
> >     As reported in the AD paper, ICRL typically emerges when the context length exceeds the episode length. Consequently, selecting an appropriate context length for the domains considered in this work **is not challenging** (since their episode lengths are **NOT very long**). In fact, the catastrophic performance of AD in our work primarily stems from its learning of a **random source algorithm**, rather than an inadequacy in context length. In addition, our method is designed to directly train on the optimal action labels (probablistically) instead of learning an improvement operator of a source algorithm. Thus, our method imposes a **more relaxed requirement on the context**. This is similar to DPT, as we discussed in the paragraph of **Decision Pretrained Transformer** on Page 4 of the manuscript. In our work, we set the context length equal to the episode length, **adhering to the setup in DPT**. We will address the confusions more clearly in our revised version of the paper.
> >
> >   [1] Laskin, M., Wang, L., Oh, J., Parisotto, E., Spencer, S., Steigerwald, R., ... & Mnih, V. (2022). In-context reinforcement learning with algorithm distillation. arXiv:2210.14215.
> >
> >
> > 6. ''*I am curious why samples beyond the N collected for computing the average reward (in Algorithm 2, step 2) are discarded. Why would extra samples be problematic? Moreover, if exactly N samples per action are needed, why is there a need to sample from a random policy at all? Wouldn’t a simple for-loop suffice?*''
> >
> > - **Response:** We thank the reviewer for highlighting this point. The "for-loop" mentioned by the reviewer is essentially **identical** to a uniform random policy in expectation. In fact, the focus of our work is **not on selecting** a (random) policy. Rather, it is on how, **given a (random) policy**, we can use it to collect useful data, particularly the optimal action labels.
> > **More importantly**, the extra samples in the MAB problem are **not problematic**. We previously discarded the samples beyond $N$ just to be consistent with Theorem 1 in the manuscript where we consider the same number of samples for $a^\star$ and other $a$ for simplicity. Nevertheless, note that Theorem 1 can be easily adapted to scenarios where **different $N$** are selected for each action in Eq.(6). This is:
> > *With probability at least $1-\delta$, it holds that
> >     \begin{align}
> >         \frac{1}{N_{a^\star}}  \sum_{i=1}^{N_{a^\star}} r_i(s_q, a^\star) \geq \max_{a \in A \setminus \{a^\star\}} \frac{1}{N_a} \sum_{i=1}^{N_a}  r_i(s_q, a),
> >     \end{align}
> >     when the trust horizon $N := \min_{a \in A} N_a$  satisfies
> >     \begin{align}
> >         N \geq \frac{8B^2}{\left(\mathbb{E}[r(s_q, a^\star)] - \max\limits_{a \in A \setminus \{a^\star\}} \mathbb{E}[r(s_q, a)] \right)^2} \log \left( \frac{1+\sqrt{1-\delta}}{\delta} \right).
> >     \end{align}*
> > **This new version of Theorem 1** as well as the complete proof will be included in the revised paper.

---

> > > ### Author Response · Authors · 2025-02-21
> > > **Response to Reviewer 4kRM (Part 3)**
> > >
> > > 7. ''*For Theorem 1, the notation is inconsistent: you sum over the index $i\in\{1, \cdots, N\},$ yet there is no corresponding index in the reward function. Additionally, the reward function is denoted as $r(s, a)$ in one instance and as $r(s, a \mid \pi)$ in another.
> > > --> Please ensure that the notation is consistent.*''
> > >
> > > - **Response:** We thank the reviewer for bringing up this issue. The reviewer is correct. In Eq.(6), we compare the empirical average reward by $a^\star$ and $a \in A \setminus \{a^\star\}$, where the reward function is **stochastic** in the MAB problem. Our **new version of Eq.(6)** (see below) will address the notation inconsistencies by incorporating a **subscript $i$** to $r$.
> > >     \begin{align}
> > >         \frac{1}{N_{a^\star}}  \sum_{i=1}^{N_{a^\star}} r_i(s_q, a^\star) \geq \max_{a \in A \setminus \{a^\star\}} \frac{1}{N_a} \sum_{i=1}^{N_a}  r_i(s_q, a),
> > >     \end{align}
> > > Previously, we used $r(s, a \mid \pi)$ to denote that the action $a$ is selected under the policy $\pi$. However, we agree that this can lead to inconsistency. Since we have already mentioned in the statement of Theorem 1 that select each arm under $\pi$, we will use $r(s_q, a)$ instead of $r(s_q, a \mid \pi)$ for consistency.
> > >
> > >
> > >
> > > 8. ''*The definition of ‘trust horizon’ is confusing due to its different interpretations in the contexts of MABs and MDPs.
> > > --> Please provide explicit definitions for ‘trust horizon’ and ‘trustworthiness’ to ensure clarity for the reader.*''
> > >
> > > - **Response:** We thank the reviewer for pointing this out. Despite different interpretations in MABs and MDPs, the **trust horizon in both cases measures the possibility that "$a^\star$ outperforms any other $a$"**. In MABs, the empirical average reward with an increasing $N$ converges to the expected reward (by the law of larger numbers), thus, as the horizon of MABs increases, the likelihood of $a^\star$ yielding average rewards larger than other actions increases. In MDPs, an increasing $N$ corresponds to increasingly negligible $\xi_a$ and $\xi_{a^\star}$ in Eq.(77) and Eq.(78), as both $\xi_a$ and $\xi_{a^\star}$ are heavily discounted by $\gamma < 1$ raised to the power of more than $N+1$. In this context, $Q_\text{MDP}^{\pi, N}(s_q, a)$ converges to $Q_\text{MDP}^\pi(s_q, a)$ as $N$ increases (same for $Q_\text{MDP}^{\pi, N}(s_q, a^*)$ and $Q_\text{MDP}^\pi(s_q, a^*)$). As a result, Eq.(11) is more likely to hold based on Assumption 2 and the law of larger numbers, thereby making $a^\star$ increasingly outstanding. That being said, we do agree with the reviewer that more explicitly explaining the trust horizon in the main text can significantly enhance our paper. In our revised version, we will provide the **formal definitions of "trust horizon" and "trustworthiness"** for the MAB and MDP respectively to ensure an improved clarity.

---

> > > > ### Author Response · Authors · 2025-02-21
> > > > **Response to Reviewer 4kRM (Part 4)**
> > > >
> > > > 9. ''*The setup for sparse rewards is not clearly defined. It is implicitly assumed to be the case in which a reward is received only upon reaching the goal. However, what about cases where a reward is obtained only twice in a trajectory? Would that not also constitute a sparse reward setting? And which action-labeling algorithm should practitioners use in such cases?*''
> > > >
> > > > - **Response:** The reviewer is correct. The sparse-reward setting in this work refers to the case where **a reward is received only upon achieving the goal**. We **present this in Appendix D.1** when introducing the environment setup. As suggested by the reviewer, we will introduce the RL environments in the main texts of our revised paper, thus including the details of the sparse-reward setting. i) To answer the reviewer's question, in a broad sense, obtaining a reward twice in the trajectory can be classified as the sparse-reward setting as the agent does not receive the reward at each step.  In general, structuring rewards to be received solely upon achieving the goal is a common practice in RL environments such as Gridworld (Darkroom) and the game of Go.
> > > > ii) We then answer the reviewer's question regarding the choice of action-labeling algorithm. When there is only one reward (for reaching the goal) in the trajectory, the action that reaches the goal using **fewest steps** (Algorithm 3) is **esentially the same as** the action that maximizes the **largest discounted cumulative reward or return** (Algorithm 5). This is because that the discounted return $\sum_t \gamma^t r_t$ has a term $\gamma^t$ where $t$ is the time step. Thus, fewer steps (smaller $t$) corresponds to larger discounted return (larger $\gamma^t$). In this work, we focus on the RL environments that have only one reward in the trajectory, and Algorithm 3 works well. However, if there is an extra reward granted for a subgoal in the trajectory, Algorithm 3 has the possibility to get stuck at the subgoal. In this context, one can **employ Algorithm 5** that selects the optimal action labels directly by the largest discounted return.
> > > >
> > > >
> > > >
> > > > 10. ''*The motivating example involving healthcare applications in the introduction appears misplaced. How is a uniform policy a reasonable choice in healthcare? Moreover, how does the healthcare example relate to Assumption 2, and can it satisfy that assumption?
> > > > --> The authors should either revise the motivating example or ensure consistency by justifying Assumption 2 as well as the use of a uniform random policy in healthcare applications.*''
> > > >
> > > > - **Response:** We thank the reviewer for bringing this to our attention.  **The healthcare example is used to represent the case where the data may not come from a complete episode**. In this case, AD is not suitable as it requires to learn from a complete RL history to learn the improvement operator. Besides, **our intention was NOT to claim that the uniform policy is reasonable in healthcare, but that the data collected in healthcare scenarios can be very limited**. Thus it could be impossible to train a good policy. In particular, as it is likely to have unlabled actions for many states, there **may be no option other than using a random policy to label** those (assuming no expert input is available).  As we mentioned in the earlier response, this work is not about selecting a policy. We explore how to effectively perform ICRL when only the random policy is available. As we discussed in the comments after Assumption 2 (and more detailed discussion in the revised version of paper), we acknowledge that Assumption 2 holds for some problems like Darkroom in this work and may not hold for some problems like healthcare. We agree with the reviewer that **it would be beneficial to revise the motivating example** to avoid the confusion. As such, we will modify this in our revised manuscript.
> > > >
> > > >
> > > > We sincerely appreciate the reviewers' insightful feedback, which will significantly improve the quality of our paper.

---

> > > > > ### Comment · Reviewer_4kRM · 2025-02-28
> > > > >
> > > > > Thank you for the responses. It also helps me to see the changes you have proposed as a revised draft because TMLR provides a way to upload revised drafts during the review process (please mark the paragraphs with changes in blue for now).

---

> > > > > > ### Author Response · Authors · 2025-02-28
> > > > > >
> > > > > > Thank you for your kind response and thoughtful suggestions. We fully agree with your advice.
> > > > > >
> > > > > > - Since *TMLR* strongly recommends that we submit the revised version of our manuscript **only after all the reviews have been released**, we will submit the revised paper as soon as possible after all the reviews have been released.
> > > > > > - We will clearly mark all the modifications in blue.

---

> > > > > > > ### Author Response · Authors · 2025-03-21
> > > > > > > **Thank you note**
> > > > > > >
> > > > > > > Dear Reviewer 4kRM,
> > > > > > >
> > > > > > > As we approach the end of the rebuttal period, we would like to express our sincere gratitude once again for your insightful comments and constructive feedback. **We hope that our responses and the revised manuscript have addressed your concerns**. If you have any additional comments, please feel free to let us know.

---

### Review · Reviewer_SRJi · 2025-03-07

**Summary Of Contributions:**

The paper "Random Policy Enables In-Context Reinforcement Learning within Trust Horizons" focuses on the pretraining of fundation decision models, that aim at solving a variety of tasks in various environnements, given a prompt specifying the context as input. Typically, the context prompt contains a set of pre-collected transitions (state-action-reward) from the environement to be solved. The training task is to learn to select the best action given a current state and a context. Concurrent approaches from the litterature usually leverage optimal - or well trained - policies to label actions corresponding to each query state, that the process attempts to imitate from the context. Authors claim that they propose the first ICRL approach based on fully random policies for collecting training data.

**Audience:**

Yes

**Broader Impact Concerns:**

.

**Claims And Evidence:**

No

**Requested Changes:**

- Improve the formalization of the problem, the presentation of the concurrent approaches and of the experimental methodology
- Further study conditions of validity of assumption 2
- I would suggest to permute algo3 and 5 (algo3 in appendix and algo5 in main), since algo5 is more easy to connect with the rest of the discussion. algo 3 coming after theorem 2 is very confusing. This makes feel that pratical algorithms are full disconnected from the theoretical analysis.
- Further analysis to connect theorem 2 with algo 5
 - Algo 3 looks extremly costly,  and can loop infinitely if start is a bit far from the goal. A random policy cannot reach any far state in a reasonnable time. Please discuss (or remove that algo which looks not convincing)
- The approach requires that we use environements that allow to reset anywhere. Please discuss.
- Additionnal results obtained with a fixed time budget common for all approaches looks mandatory.
- Sentence :  "It shows that the accuracy monotically decreases with N, which ideally should lead to... ". Why ideally ? I would rather say fortunately... And why this is not the case ?
- From my point of view theorem 1 is already well known and given in many bandit papers. Also, for the MAB problem, it looks natural that it works, while using UCB policies could look better to collect good data. I would suggest to focus on MDP, which is clearly more interesting (but for which the approach looks rather limitative).




Minor :
 - in (6), r is given depending on \pi. why ? It is not the case elsewhere...

**Strengths And Weaknesses:**

Strengths:
   - Interesting and useful problem
   - Theoretical guarantees (while using somehow irrealistic assumptions)

Weakness:
   - Presentation/Formalization of the problem difficult to follow. The problem is not well presented, which complicates understanding of what follows. Finally, I had to refer to the formalization from the DIT paper to understand the problem setting. In particular, what is C is not very clear and how we use the methods at test time is not discussed. Also, I felt very confusing to avoid distinguishing the behavior policy (that does not need to be optimal in concurrent approaches), from the action labeller (for which we require good accuracy). The introduction of $\Delta$ in the formalization, and the notion of action label, are also particularly unclear.
   - Theoretical assumption 2 is far from being realistic. I appreciated that authors studied its validity for a very simple minigrid problem, but even for a straight minigrid with two states with rewards (the highest the most far from start), I feel it does not hold yet, right ? The random policy will indicate going in the direction of the low but close reward... It would be useful that authors study conditions of validity to complement the sudy.
  - Theoretical study indicates that with an horizon sufficiently long, the estimation of Q from a random policy is accurate. But it really looks counter-intuitive, or at least additionnal assumptions are missing to me, especially when authors set Nep=1 in their algorithms. Please clarify (I suppose that Nep is recovered by considering a high number of weak labels during training, which explains the accuracy loss observed in fig 5b, but I feel that  authors should give something more to connect everything)
 - Finally, the approach comes down to replace the need for having a well trained labeller by a huge dataset collected at random. Isn't this at least as constraining ?  I feel this is very costly. I cannot find any place where authors give the size of the collected dataset. Authors give only a quick result at the end of appendix (additional results) regarding computing ressources, while it is core for analysis the interest of the approach. The average time given for MDP looks significantly greater as concurrent approaches. And it is given only for the sparse problem, for which authors use expert knowledge to start the random policy close to the goal (from my point of view, this trick looks cheating, as getting this knowledge is close to requiring the optimal). At least, authors should consider to give performance metrics for a given time budget, including everything (dataset collection + training), common for every approach. With a infinite budget, it looks obvious that we can reach better estimations for small problems.
- Experimental details are missing. For instance, how is set the context for the offline experiment ?

---

> ### Author Response · Authors · 2025-03-11
> **Response to Reviewer SRJi (Part 1)**
>
> We sincerely appreciate the reviewer’s thoughtful suggestions and detailed feedback. We are pleased to hear that the reviewer finds our work interesting and useful, and recognizes our theoretical guarantees.
>
> Furthermore, we believe that the concerns and requested changes outlined by the reviewer will significantly strengthen our paper. Below, we provide detailed responses to each comment and requested change from the reviewer.
>
> 1. ''*Presentation/Formalization of the problem difficult to follow. The problem is not well presented, which complicates understanding of what follows. Finally, I had to refer to the formalization from the DIT paper to understand the problem setting. In particular, what is C is not very clear and how we use the methods at test time is not discussed. Also, I felt very confusing to avoid distinguishing the behavior policy (that does not need to be optimal in concurrent approaches), from the action labeller (for which we require good accuracy). The introduction of $\Delta$ in the formalization, and the notion of action label, are also particularly unclear.
> --> Improve the formalization of the problem, the presentation of the concurrent approaches and of the experimental methodology*''
>
> - **Response:** We thank the reviewer for pointing this out.
> **i)** As the reviewer noted in the *Summary Of Contributions*, *C* (context) is presented by a set of collected transitions (state-action-reward-next_state) from each pretraining environment. In this work, since we focus on scenarios where only the random policy is accessible, we consider **the context *C* to be collected by interacting with pretraining environments under the random policy**. The reviewer is kindly referred to Algorithm 1 for details. We will introduce the context *C* with more details (e.g., referring to Algorithm 1) in Section 2.2 of our revised manuscript. In addition, this work focuses on the pretraining dataset generation of ICRL, thus following identical training and test processes as that of AD and DPT. That being said, we still provide a **pseudo-code for pretraining and deployment processes in Algorithm 6**. In particular, there are two types of deployments/tests: online and offline. In online deployment, the pretrained model collects its own context by iteratively interacting with the test environments. In offline deployment, the pretrained model directly uses a sampled context from an offline context dataset, which is pre-collected using the random policy to interact with the test environments. **We will explain more details about these experimental setup in the main texts** in our revised manuscript.
> **ii)** Note that the baseline methods, such as AD, DIT, DPT, label the actions directly using the well-trained learning history or optimal policies. Thus, they require well-trained/optimal policies for **all pretraining environments**, which needs **an ocean of data** to train over. In the example of *Darkroom* problem, it requires **2,401,000,000 transitions** to obtain well-trained policies. We will discuss this in more details later in the 6th point of this response. Since only the random policy is accessible in this work, the baseline methods will directly label the action using the random policy. In our method, we introduce "action-labeling algorithms", which are **NOT policies but rather techniques** for utilizing the random policy to label the actions (our algorithms 2, 3, 5). We will also show in the 6th point of this response that the cost of our method is much smaller than training well-trained policies.
> **iii)** We use $\Delta$ to denote the probability distribution over the action space following the ICRL literature[1-2]. The action label $a_l$ denotes the prediction (output) of the FM given the context $C$ and query state $s_q$. However, we **agree with the reviewer** that these notations should be introduced more clearly in the formalization. We will do that in our revised version of the manuscript, as well as **more clearly presenting the current ICRL methods and our experimental settings**.
>
>   [1] Lee, J., Xie, A., Pacchiano, A., Chandak, Y., Finn, C., Nachum, O., & Brunskill, E. (2024). Supervised pretraining can learn in-context reinforcement learning. NeurIPS 2023.
>
>   [2] Dong, J., Guo, M., Fang, E. X., Yang, Z., & Tarokh, V. In-Context Reinforcement Learning Without Optimal Action Labels. ICML 2024 Workshop on In-Context Learning.
>
>
> 2. ''*I would suggest to permute algo3 and 5 (algo3 in appendix and algo5 in main), since algo5 is more easy to connect with the rest of the discussion. algo 3 coming after theorem 2 is very confusing. This makes feel that pratical algorithms are full disconnected from the theoretical analysis.*''
>
> - **Response:** We thank the reviewer for bringing this to our attention. We agree with the reviewer that **permuting algorithms 3 and 5 can connect the theretical analysis and the practical algorithms better**. We will do that in our new version of paper.

---

> > ### Author Response · Authors · 2025-03-11
> > **Response to Reviewer SRJi (Part 2)**
> >
> > 3. ''*Theoretical assumption 2 is far from being realistic. I appreciated that authors studied its validity for a very simple minigrid problem, but even for a straight minigrid with two states with rewards (the highest the most far from start), I feel it does not hold yet, right ? The random policy will indicate going in the direction of the low but close reward... It would be useful that authors study conditions of validity to complement the study.
> > -->Further study conditions of validity of assumption 2*''
> >
> > - **Response:** Let us break down very good points the reviewer is bringing up in three different items.
> > **i)**  It is true that the validity of Assumption 2 **depends on the specific problem**, and is related to e.g., **rewards, discount factor $\gamma$**, etc. However, **Assumption 2 can be more general than one might initially think it is**, as our Assumption 2 does not state that $Q_{MDP}^\pi(s, a)$ ($\pi$ is the random policy) is equivalent to $Q_{MDP}^\star(s, a)$; rather, it assumes that **they share the same action maximizer**. We next give two examples using the MDP proposed by the reviewer in which **Assumption 2 holds**.
> > a) We consider the same setting as in Figure 9 (with $\gamma = 0.99$) except assigning an additional reward of 0.1 for being in the state $s_4$. In this case, **the action that maximizes the Q function for both policies is the same**, i.e., $a_0=\arg\max_a \,  Q_{MDP}^\pi(s,a) =\arg\max_a \, Q_{MDP}^\star(s,a)$ for all states as shown below
> >
> > | $Q_{MDP}^\pi(s, a)$ | $a_0$ | $a_1$ |
> > |----------|----------|----------|
> > | $s_0$   | **24.169**  | 22.637  |
> > | $s_1$   | **24.169**  | 21.562  |
> > | $s_2$   | **22.637**  | 20.923  |
> > | $s_3$   | **21.562**  | 20.707  |
> > | $s_4$   | **20.923**  | 20.707  |
> >
> > | $Q_{MDP}^\star(s, a)$ | $a_0$ | $a_1$ |
> > |----------|----------|----------|
> > | $s_0$   | **99.999**  |  98.999 |
> > | $s_1$   | **99.999**  |  98.009 |
> > | $s_2$   | **98.999**  |  97.029 |
> > | $s_3$   | **98.009**  |  96.159 |
> > | $s_4$   | **97.029**  |  96.159 |
> >
> > b) We consider the same setting as in Figure 9 (with $\gamma = 0.8$) except assigning an additional reward of 0.9 for being in the state $s_4$. In this case, the random policy selects $a_0$ at the states $s_0, s_1, s_2$ and $a_1$ at the states $s_3, s_4$ (see $Q_{MDP}^\pi(s, a)$ below). This is noted by the reviewer. However, please also notice that **the optimal policy selects exactly the same optimal actions as that of the random policy**, i.e., $a_0$ for states $s_0, s_1, s_2$ and $a_1$ for states $s_3, s_4$ (see $Q_{MDP}^\star(s, a)$ below), thus **Assumption 2 still holding**.
> >
> > | $Q_{MDP}^\pi(s, a)$ | $a_0$ | $a_1$ |
> > |----------|----------|----------|
> > | $s_0$   | **2.718**  | 1.577  |
> > | $s_1$   | **2.718**  | 1.225  |
> > | $s_2$   | **1.577**  | 1.486  |
> > | $s_3$   | 1.225  | **2.491**  |
> > | $s_4$   | 1.486  | **2.491**  |
> >
> > | $Q_{MDP}^\star(s, a)$ | $a_0$ | $a_1$ |
> > |----------|----------|----------|
> > | $s_0$   | **4.999**  |  3.999 |
> > | $s_1$   | **4.999**  | 3.199  |
> > | $s_2$   | **3.999**  | 3.599  |
> > | $s_3$   |  3.199 | **4.499**  |
> > | $s_4$   |  3.599 | **4.499**  |
> >
> > In both cases above, the random policy and the optimal policy have the same optimal action (action maximizer), despite having different Q-functions. This is precisely what Assumption 2 conveys. We will include a discussion of these new examples above in our revised manuscript.
> > **ii)** More importantly, **we don't claim that we can use a random policy to solve all problems**. As we have discussed (and will discuss more in the revised paper) right after Assumption 2, **we acknowledge that Assumption 2 will not hold for all MDPs**. Indeed, it is **impossible** that a random policy and the optimal policy select the same optimal actions for **all problems**. However, **Assumption 2 does hold in the tasks of interest in this work**: gridworld navigation tasks where there is **only one reward** assigned upon reaching the goal, which are **commonly considered as the benchmark environments** in ICRL literature. And as demonstrated in the previous point, the assumption may result more general than this. We have provided the **theoretical and experimental proofs** for Assumption 2 in Appendix B.2. In general, structuring rewards to be received solely upon achieving the goal is a common practice in RL environments such as Darkroom, the game of Go, Chess, etc.
> > **iii)** Finally, we **do agree with the reviewer** that it will be beneficial to more explicitly mention the conditions for the validity of Assumption 2. We will **make it more clear** in the revised version that we focus on the gridworld navigation benchmark environments that **assign a reward only for achieving the goal**, for which we have provided the theoretical and experimental proofs.

---

> > > ### Author Response · Authors · 2025-03-12
> > > **Response to Reviewer SRJi (Part 3)**
> > >
> > > 4. ''*Theoretical study indicates that with an horizon sufficiently long, the estimation of Q from a random policy is accurate. But it really looks counter-intuitive, or at least additionnal assumptions are missing to me, especially when authors set Nep=1 in their algorithms. Please clarify (I suppose that Nep is recovered by considering a high number of weak labels during training, which explains the accuracy loss observed in fig 5b, but I feel that authors should give something more to connect everything)*''
> > >
> > > - **Response:** We thank the reviewer for highlighting this point.
> > > **i)** Please note that Theorem 2 is based on Assumption 2 that considers **infinite-horizon Q-function** and **not the stochastic estimates** of the Q-function which could lead to larger variance for larger $N$. With this in mind, in Eq.(77) and Eq.(78), we split the infinite-horizon Q-function into two parts: $Q_\text{MDP}^{\pi, N}(s_q, a)$ and $\xi_a$ as follows (same for $Q_\text{MDP}^{\pi, N}(s_q, a^*)$ and $\xi_{a^*}$)
> > > $$Q_\text{MDP}^{\pi, N}(s_q, a) = \mathbb{E} \left[ \sum_{t=0}^{N} \gamma^t r(s_t, a_t) \mid s_0=s_q, a_0=a \right]$$
> > > $$\xi_a = \mathbb{E} \left[ \sum_{t=N+1}^{\infty} \gamma^t r(s_t, a_t) \mid s_0=s_q, a_0=a \right]$$
> > > From the equations above, an **increasing $N$** corresponds to **increasingly negligible** $\xi_a$ and $\xi_{a^\star}$, as both $\xi_a$ and $\xi_{a^\star}$ are discounted by $\gamma < 1$ to the power of a number larger than $N+1$. In this context, $Q_\text{MDP}^{\pi, N}(s_q, a)$ converges to $Q_\text{MDP}^\pi(s_q, a)$ as $N$ increases (same for $Q_\text{MDP}^{\pi, N}(s_q, a^*)$ and $Q_\text{MDP}^\pi(s_q, a^*)$). As a result, Eq.(11) is more likely to **hold based on Assumption 2 and the law of larger numbers**, thereby making $a^\star$ increasingly outstanding. Intuitively, **a larger $N$ corresponds to a closer approximation of the infinite-horizon MDP**, where the optimal actions emerge under the random policy (by Assumption 2). **We have discussed this right after Theorem 2** briefly due to the space constraint. We will **include more discussions and details** in our revised manuscript.
> > > **ii)** On the other hand, while a larger $N$ **in practice** may lead to increased variance, setting $N$ too small can make it difficult to distinguish actions for states far from the goal. For instance, if $N=1$ and no actions for a given state $s_0$ can reach the goal (i.e., no rewards received), then $Q_{MDP}^{\pi, N}(s_0, a) = 0$ for any action $a$ at $s_0$, making all actions appear **equally good**. In this sense, it is intuitive to consider a sufficiently large $N$ so that the states far from the goal can navigate towards it, thereby enabling the identification of an outstanding action. We will introduce the trade-off of $N$ in practice with more details in the following point.
> > > **iii)** Please note that $N_{ep}$ is the number of episodes, and setting $N_{ep}=1$ is a general practice for sample efficiency. Moreover, it is worth noting that Theorem 2 considers the **worst-case guarantee of selecting the optimal actions over ALL states** in the state space. It is therefore could be **conservative**. However, in practice, we only select the state-action pairs that can achieve the goal within $N$ steps, because it is unlikely to distinguish the actions withouting reaching the goal as we mentioned in the previous point. We thus only record the accuracy of selecting the optimal actions for those states that can reach the goal (Figure 5(b)). In this sense, there is a trade-off for selecting the trust horizon $N$. A larger $N$ can train on more states but with lower accuracy (for selected states, not ALL states) for selecting the optimal actions (see Figure 5(b)), while a smaller $N$, despite with higher accuracy, may induce the partially short-sighted training of the FM (see Figure 7). This trade-off in practice has been validated by our experimental results in Figure 6.
> > >
> > >
> > >
> > > 5. "*Further analysis to connect theorem 2 with algo 5*"
> > >
> > > - **Response:** We appreciate the reviewer's advice. Similar to the connections between Theorem 1 and algorithm 2, Theorem 2 implies that the action with the **highest average reward** under the random policy is, probabilistically, the optimal action. Despite being conservative, Theorem 2 hints to the fact that one should select the action that yields the highest average reward in practice (Algorithm 5). We set $N_{ep}=1$ to maintain the sample efficiency. Empirically, we find that this selection is enough to attain superior performance. While selecting $N_{ep}>1$ is an option, it will come at a higher computational cost. We will make these connections better in our revised manuscript.

---

> > > > ### Author Response · Authors · 2025-03-12
> > > > **Response to Reviewer SRJi (Part 4)**
> > > >
> > > > 6. ''*Finally, the approach comes down to replace the need for having a well trained labeller by a huge dataset collected at random. Isn't this at least as constraining ? I feel this is very costly. I cannot find any place where authors give the size of the collected dataset. Authors give only a quick result at the end of appendix (additional results) regarding computing ressources, while it is core for analysis the interest of the approach. The average time given for MDP looks significantly greater as concurrent approaches. And it is given only for the sparse problem, for which authors use expert knowledge to start the random policy close to the goal (from my point of view, this trick looks cheating, as getting this knowledge is close to requiring the optimal). At least, authors should consider to give performance metrics for a given time budget, including everything (dataset collection + training), common for every approach. With a infinite budget, it looks obvious that we can reach better estimations for small problems.
> > > > --> Algo 3 looks extremly costly, and can loop infinitely if start is a bit far from the goal. A random policy cannot reach any far state in a reasonnable time. Please discuss (or remove that algo which looks not convincing)
> > > > --> Additionnal results obtained with a fixed time budget common for all approaches looks mandatory.*''
> > > >
> > > > - **Response:** We thank the reviewer for the thoughtful comments, however we **respectfully disagree** with some of the assessements.
> > > > **i)** Note that the baseline methods such as AD, DIT, DPT require well-trained or optimal policies for **ALL pretraining environments**, which could take a large amount of data. In contrast, our approach requires **ONE random policy** only. Let us consider *Darkroom* as an example, obtaining well-trained policies for all pretraining environments in the baseline methods will need **2,401,000,000 transition data**, while our method uses **21,445,147 transition data** only. In the revised version of our paper, we **will include a table that explicitly demonstrates this comparison**.
> > > > **ii)** Please also notice that our Algorithm 3 is **neither infinite budget nor extremely costly**. Let us use our method with $N=7$ as an example which achieves the best performance as shown in our ablation study. Our method with $N=7$ **uses 21,445,147 transition data only and consumes 82 seconds only** in the dataset generation. In contrast, obtaining well-trained policies for all pretraining environments in the baseline methods **uses 2,401,000,000 transition data and consumes 3452 seconds** in the dataset generation. Furthermore, the baseline methods need to **learn and maintain policy and value networks** for all pretraining environments **(additional costs)**, which is not the case in our method since the only thing we do is collecting at random. It is true that the baseline methods take about 20-40 seconds if they just employ the random policy (no well-trained policies). But they demonstrate a **catastrophic performance** as shown in Figures 2 and 3.
> > > > **iii)** Let us clarify possible misunderstandings which we will make more clear in the new version of our manuscript. We **do NOT use any knowledge** to start the policy (or sample query states). We **always sample the query state randomly**. If there is no action for a query state that can reach the goal within $N$ steps, we **randomly** sample another query state. Since it is randomly sampled (i.e., NOT always far from the goal), it **will not loop forever. Our experiments validate this.** Even for small trust horizons like $N=1, 3, 7$, it takes only 50s, 72s and 82s to generate the dataset respectively. **We never loop infinitely**.
> > > > **iv)** As we mentioned in the manuscript and our earlier response, we focus on the pretraining dataset generation of ICRL, therefore maintaining an identical training process as that of the baseline methods. However, we **do agree with the reviewer** that it is worth comparing our method with the baselines under **the same dataset generation time budget**. It would be even more insightful and more interesting to examine how our method performs under **a reduced time budget**. Our preliminary experiments demonstrate that, despite with degraded performance and increased vairance compared to before, our SAD method still **outperforms all baselines under the same time budget** (38 seconds, which is consumed by DIT), especially in the offline deployment. This implies that our method has a **more robust performance than the baselines** when tackling the random contexts, as the offline deployment uses pre-collected random contexts. We will include it in our new version of the manuscrpit.

---

> > > > > ### Author Response · Authors · 2025-03-12
> > > > > **Response to Reviewer SRJi (Part 5)**
> > > > >
> > > > > 7. ''*Experimental details are missing. For instance, how is set the context for the offline experiment ?*''
> > > > >
> > > > > - **Response:** We thank the reviewer for pointing this out. Note that this work focuses on the pretraining dataset generation, and we thus consider identical training and test processes as that of AD and DPT. We comment this in the Experiments section and provide the citations to the baseline work. In addition, we provide a pseudo-code for pretraining and deployment in Appendix C (see Algorithm 6). In particular, the context in the offline deployment is collected by **using the random policy to interact with the test environments**. We will introduce this with more details in our new version of paper.
> > > > >
> > > > >
> > > > >
> > > > >
> > > > > 8. ''*The approach requires that we use environements that allow to reset anywhere. Please discuss.*''
> > > > >
> > > > > - **Response:** We believe the reviewer referring to the MDP problems, as the MAB problems do not need to reset the state. In our experiments, the benchmark environments we considered are **identical** to that in AD, DIT and DPT. We **do not have any extra requirements or modifications on the envrironments**. On the other hand, random initialization is a **general technique** to guarantee the robustness of the algorithm, which has been employed in the current ICRL baselines (AD, DIT, DPT). It is **not specifically "required" by our method**.
> > > > >
> > > > >
> > > > >
> > > > > 9. ''*Sentence : "It shows that the accuracy monotically decreases with N, which ideally should lead to... ". Why ideally ? I would rather say fortunately... And why this is not the case ?*''
> > > > >
> > > > > - **Response:** Thank you for bringing this to our attention. We were intending to comment that **at first glance** a lower accuracy of selecting the optimal action will lead to lower performance. But as we explain right after that sentence, "a large trust horizon $N$ in the MDP leads to the low accuracy of selecting the optimal action, whereas a small $N$ may induce the partially short-sighted training of the FM, as we solely train on the states at most $N$ steps from the goal, instead of all states (refer to Figure 7)". We will **enchance the language and improve the clarity** in the revised manuscript.
> > > > >
> > > > >
> > > > >
> > > > > 10. ''*From my point of view theorem 1 is already well known and given in many bandit papers. Also, for the MAB problem, it looks natural that it works, while using UCB policies could look better to collect good data. I would suggest to focus on MDP, which is clearly more interesting (but for which the approach looks rather limitative).*''
> > > > >
> > > > > - **Response:** Thank you for your thoughtful comments. Please note that our problem of interest in this work considers the scenarios that **only the random policy** is accessible, therefore UCB not applicable. We **agree with the reviewer** that the theoretical analysis for the MAB problems is **natural**, and more importantly it is **a very intuitive example** that the random policy could work **under some conditions**. But we **also agree** with the reviewer that MDP is more complicated and more interesting. We will **condense the MAB part appropriately and focus more on the MDP** in our new version of the paper.
> > > > >
> > > > >
> > > > >
> > > > > 11. ''*in (6), r is given depending on $\pi$. why? It is not the case elsewhere...*''
> > > > >
> > > > > - **Response:** Thank you for highlighting this point. We intended to use $r(s_q, a \mid \pi)$ to denote that the action $a$ is selected under the policy $\pi$. However, we agree that this can lead to misunderstandings. Since we have already mentioned in the statement of Theorem 1 that select each arm under $\pi$, we will just use $r(s_q, a)$ instead of $r(s_q, a \mid \pi)$ in our revised paper.

---

> > > > > > ### Author Response · Authors · 2025-03-12
> > > > > > **Response to Reviewer SRJi (Part 6)**
> > > > > >
> > > > > > Finally, we would like to summarize and emphasize our problem of interest, contributions and clarifications for the reviewer. We will also make them more clear in our revised paper.
> > > > > > - The current ICRL methods like AD, DIT, DPT will demand varying-degree well-trained or even optimal policies for **all pretraining environments**, which is **much more expensive than just collecting under random policies** (as we discussed in the **6th point** of this response). In some scenarios where the data is very limited and with poor quality, training such policies is even intractable, and there may be no option other than using a random policy. Therefore, understanding how ICRL works under the random policy is a crucial direction, which is the **focus of this work**.
> > > > > > - We **do acknowledge that it is too ambious and impossible to employ a random policy to solve all problems**. However, the random policy works well to figure out an action labeller in the **gridworld navigation tasks where the reward is received only upon achieving the goal**, which are commonly-considered benchmark environments in the ICRL literature. Moreover, as we discussed in the **3rd point** of this response, **Assumption 2 can be more general than one might initially think it is**, as our Assumption 2 does not state that $Q_{MDP}^\pi(s, a)$ ($\pi$ is the random policy) is equivalent to $Q_{MDP}^\star(s, a)$; rather, it assumes that **they share the same action maximizer**.
> > > > > > - **Without the need to train an ocean of data** to obtain the optimal policies for all pretraining environments, our method surpasses all baseline approaches and achieves a comparable performance with that of $DPT^\star$ in MAB and Darkroom-type problems, which employs the optimal policies.
> > > > > > - We **do not require any knowledge** to sample the query state (or start the policy). It is **completely randomly sampled**.
> > > > > > - Our method **never loops infinitely**, and it is **NOT very costly** especially compared to training an ocean of data to obtain the well-trained policies for all pretraining environments (refer to the  **6th point** of this response).
> > > > > >
> > > > > >
> > > > > > Once again, we **deeply appreciate the reviewer's insightful and detailed feedback**, which provides the guidance for significantly enhancing our manuscript.

---

> > > ### Comment · Reviewer_SRJi · 2025-03-18
> > >
> > > Thanks for your detailed answer
> > > But sorry but I still feel that you took a specific setting that works for you, but even with straigths lined minigrid (which is the simplest env we can find, it is easy to find settings for which assumption 2 does not hold)
> > >
> > > Please for instance consider a setting with a straight line of states s0 to s5
> > > Arriving at state s0 gets a reward of 1, s5 gets a reward of 0.9
> > > action left goes toward s0, right toward s5
> > >
> > >
> > > We take a $\gamma=0.97$
> > > Consider the values for the state s4
> > > For the random policy, I get Q(s4,left) converges to about 16,85603273 while Q(s4,right) about 17,38883125. Thus for the random policy the best action regarding Q at s4 is right. Now it is clear that the optimal policy (equal to the policy induced by $Q^*$) is going left, as $0.97^3 > 0.9$
> > >
> > > I think authors should better analyze the conditions of the assumption 2 validity in a more general perspective, or if not possible, they should remove the semi-theoretical justification about  the validity of assumption 2 in such a very specific setting, which I feel is misleading.
> > >
> > > What do you think ?
> > >
> > > Regards

---

> ### Author Response · Authors · 2025-03-18
> **Response to Reviewer SRJi**
>
> **We sincerely thank the reviewer for the continued feedback**. We appreciate the example provided. Please note that **our intention was not to claim that Assumption 2 holds for all grid world navigation problems with two sparse rewards**. We intended to convey that **Assumption 2 can be more general than one might initially think it is**. More specifically, **Assumption 2 does hold for some (but NOT ALL)** navigation problems that have two sparse rewards. In fact, we explicitly mentioned in the 3rd point of our response that **the validity of Assumption 2 depends on the specific problem, and is influenced by the factors such as rewards, discount factor $\gamma$, etc**. In our revised manuscript, we have **exactly included the same example (see below) that the reviewer just mentioned**, where Assumption 2 does not hold.
>
> "*We consider the same setting as in Figure 9 (with $\gamma = 0.97$) except assigning an additional reward of 0.9 for being in the state $s_4$. The convergent Q-tables below indicate that under the random policy $\pi$, the optimal action for states $s_3$ and $s_4$ is $a_1$, whereas under the optimal policy, it is $a_0$. This discrepancy indicates that Assumption 2 does not hold.*"
>
> | $Q_{MDP}^\pi(s, a)$ | $a_0$ | $a_1$ |
> |----------|----------|----------|
> | $s_0$   | **13.596**  | 12.375  |
> | $s_1$   | **13.596**  |  11.920 |
> | $s_2$   | **12.375**  | 12.202  |
> | $s_3$   |  11.920 | **13.238**  |
> | $s_4$   |  12.202 |  **13.238** |
>
> | $Q_{MDP}^\star(s, a)$ | $a_0$ | $a_1$ |
> |----------|----------|----------|
> | $s_0$   | **33.333**  |  32.333 |
> | $s_1$   | **33.333**  |  31.363 |
> | $s_2$   | **32.333**  |  30.422 |
> | $s_3$   | **31.363**  |  30.409 |
> | $s_4$   | **30.422**  |  30.409 |
>
>
>
> With the reorganization of the paper, we make explicit that **"our applicable domains in this work are the grid world navigation problems where the reward is received only upon reaching the unique goal (i.e., there is only one reward granted)"**. The commonly-considered benchmark problems in the ICRL literature[1-3] that we use in this work fall into this category. **Assumption 2 holds for these problems, and Proposition 1 provides the necessary mathematical justification**. Accordingly, **we believe that retaining Proposition 1 is essential to ensuring the rigor of Assumption 2’s validity** for the grid world navigation problems with a single sparse reward, and **we believe it is valuable for readers**.
>
>
> Additionally, we will present empirical evidence for the grid world with two sparse rewards, **demonstrating that the validity of Assumption 2 depends on problem-specific factors such as rewards and the discount factor $\gamma$** (as illustrated in the example above). However, **it is crucial to highlight that none of our experimental environments fall into this category.**
>
>   [1] Lee, J., Xie, A., Pacchiano, A., Chandak, Y., Finn, C., Nachum, O., & Brunskill, E. (2024). Supervised pretraining can learn in-context reinforcement learning. Advances in Neural Information Processing Systems, 36.
>
>   [2] Dong, J., Guo, M., Fang, E. X., Yang, Z., & Tarokh, V. In-Context Reinforcement Learning Without Optimal Action Labels. In ICML 2024 Workshop on In-Context Learning.
>
>   [3] Laskin, M., Wang, L., Oh, J., Parisotto, E., Spencer, S., Steigerwald, R., ... & Mnih, V. (2022). In-context reinforcement learning with algorithm distillation. arXiv preprint arXiv:2210.14215.

---

> > ### Author Response · Authors · 2025-03-21
> > **Thank you note**
> >
> > Dear Reviewer SRJi,
> >
> > As we approach the end of the rebuttal period, we would like to express our sincere gratitude once again for your insightful comments and constructive feedback. **We hope that our responses and the revised manuscript have addressed your concerns**. If you have any additional comments, please feel free to let us know.

---

### Author Response · Authors · 2025-03-20
**The revised manuscript has been submitted**

Dear Reviewers and AE,

We have submitted the revised version of our manuscript based on the reviewers' comments. Please do not hesitate to reach out if you have any new comments/questions. **We sincerely appreciate your time and effort in the review process.**

Kindly note that all references to figures, equations, theorems, sections, etc., in our rebuttal correspond to the original manuscript.

---

### Decision · Action_Editor_6taJ · 2025-04-19

**Recommendation:** Accept as is

**Comment:**

The recommendation is based on the reviewers' comments, the action editor's evaluation, and the authors’ response.

This paper proposes State-Action Distillation (SAD) for in-context reinforcement learning (ICRL), which can generate an effective pretraining dataset guided solely by random policies. SAD shows notable performance improvements in ICRL benchmarks. All reviewers find the studied setting novel and the results provide new insights. The authors’ rebuttal has successfully addressed the major concerns of reviewers. Therefore, I recommend acceptance of this submission. I also expect the authors to include the new results and suggested changes during the rebuttal phase to the final version.

**Audience:**

Of broad interest to reinforcement learning and foundation models

**Claims And Evidence:**

The claims of improved in-context reinforcement learning (ICRL) are supported by the empirical results and theoretical analysis.

---

> ### Author Response · Authors · 2025-04-30
> **Camera-Ready Version**
>
> Dear AE and reviewers,
>
> We would like to express our sincere gratitude for the time and effort you have devoted to the review process. The reviewers’ thoughtful and constructive suggestions have been valuable in improving our paper. We have carefully revised the manuscript in accordance with the feedback received and have now submitted the camera-ready version.